# Phosphatidylinositol 3-phosphate and Hsp70 protect *Plasmodium falciparum* from heat-induced cell death

**Kuan-Yi Lu[1,2], Charisse Flerida A Pasaje[3], Tamanna Srivastava[2], David R Loiselle[4], Jacquin C Niles[3], Emily Derbyshire[1,2]***

[1]Department of Molecular Genetics and Microbiology, School of Medicine, Duke University, Durham, United States; [2]Department of Chemistry, Duke University, Durham, United States; [3]Department of Biological Engineering, Massachusetts Institute of Technology, Cambridge, United States; [4]Department of Pharmacology and Cancer Biology, School of Medicine, Duke University, Durham, United States

**Abstract** Phosphatidylinositol 3-phosphate (PI(3)P) levels in *Plasmodium falciparum* correlate with tolerance to cellular stresses caused by artemisinin and environmental factors. However, PI(3)P function during the *Plasmodium* stress response was unknown. Here, we used PI3K inhibitors and antimalarial agents to examine the importance of PI(3)P under thermal conditions recapitulating malarial fever. Live cell microscopy using chemical and genetic reporters revealed that PI(3)P stabilizes the digestive vacuole (DV) under heat stress. We demonstrate that heat-induced DV destabilization in PI(3)P-deficient *P. falciparum* precedes cell death and is reversible after withdrawal of the stress condition and the PI3K inhibitor. A chemoproteomic approach identified PfHsp70-1 as a PI(3)P-binding protein. An Hsp70 inhibitor and knockdown of PfHsp70-1 phenocopy PI(3)P-deficient parasites under heat shock. Furthermore, PfHsp70-1 downregulation hypersensitizes parasites to heat shock and PI3K inhibitors. Our findings underscore a mechanistic link between PI(3)P and PfHsp70-1 and present a novel PI(3)P function in DV stabilization during heat stress.

*For correspondence:
emily.derbyshire@duke.edu

**Competing interests:** The authors declare that no competing interests exist.

## Introduction

*Plasmodium* parasites are obligate intracellular pathogens that cause malaria after being transmitted to vertebrates by *Anopheles* mosquitoes. During their complex life cycle, the parasites encounter many cellular stresses as they alternate between distinct hosts and adapt to different microenvironments for successful invasion, development and replication. Febrile temperatures encountered during blood stage infection are perhaps among the most hostile stress stimuli these parasites experience. During this period, parasites progress through the ring (early), trophozoite (mid) and schizont (late) stages to produce numerous daughter merozoites capable of further red blood cell (RBC) invasion (*Kwiatkowski, 1989*; *Porter et al., 2008*; *Engelbrecht and Coetzer, 2013*). Such heat stress in many organisms, including *Plasmodium,* can induce protein denaturation and proteotoxicity, which leads to increased oxygen consumption and oxidative damage to cellular components, with prolonged exposure (*Engelbrecht and Coetzer, 2013*; *Morano et al., 2012*; *Ritchie et al., 1994*; *Roti Roti, 2008*; *Oakley et al., 2007*). Although the process by which *Plasmodium* copes with heat stress is unclear, a highly coordinated stress response is likely required to ensure their survival and replication under these conditions.

Among the human-infective *Plasmodium* species, *P. falciparum* accounts for the greatest mortality and spreading resistance to first-line artemisinin-based combination therapy jeopardizes the effectiveness of current malaria control efforts. This challenge highlights a pressing need to identify

new parasite vulnerabilities, perhaps by disrupting their ability to tolerate stress. Previous studies have demonstrated that *P. falciparum* at the ring stage is more refractory to heat stress when compared to trophozoite and schizont stages (*Kwiatkowski, 1989*; *Porter et al., 2008*; *Engelbrecht and Coetzer, 2013*). However, cyclical fever in patients with *P. falciparum* malaria often reaches 39–41°C and persists until the early schizont stage (*Crutcher and Hoffman, 1996*; *Neva and Brown, 1996*). This prolonged febrile state suggests that trophozoites and early schizonts are frequently exposed to heat shock in vivo and have likely evolved mechanisms to cope with heat stress. While details of stress response pathways in *P. falciparum* remain obscure, there is a greater understanding of the artemisinin-induced chemical stress response.

Artemisinin and its derivatives exert their antimalarial activity by generating carbon-centered radicals that cause oxidative stress and subsequent protein alkylation (*Tilley et al., 2016*; *Paloque et al., 2016*). Accumulation of alkylated proteins increases proteotoxic stress in parasites, causing a phenotype reminiscent of that induced by heat shock (*Morano et al., 2012*; *Ritchie et al., 1994*; *Roti Roti, 2008*). Increased artemisinin resistance has been found in *P. falciparum* parasites with Pfkelch13 mutations (*Miotto et al., 2015*; *Ariey et al., 2014*; *Ghorbal et al., 2014*; *Straimer et al., 2015*; *Mbengue et al., 2015*). A previous study found that PfKelch13 could modulate the level of a signaling molecule phosphatidylinositol 3-phosphate (PI(3)P) through interaction with PfPI3K (*Mbengue et al., 2015*), while other studies did not detect the interaction between PfKelch13 and PfPI3K (*Siddiqui et al., 2020*; *Gnädig et al., 2020*; *Birnbaum et al., 2020*). PfKelch13 mutations have been linked to the accumulation of PI(3)P in *P. falciparum*-infected RBCs and this increased PI(3)P level is highly correlated with parasite resistance to artemisinin (*Mbengue et al., 2015*). Therefore, PI(3)P levels could be influenced by PfKelch13 directly or PfKelch13 could indirectly influence PI(3)P levels by a mechanism independent of PfPI3K binding. Increasing PI(3)P levels by ectopically expressing a human PI3K, Vps34, in *P. falciparum* confers similar resistance (*Mbengue et al., 2015*). Intriguingly, a phenotypic screen using *piggyBac* mutagenesis showed a reduced heat tolerance in a *pfkelch13*-upregulated *P. falciparum* mutant (*Thomas et al., 2016*). If Pfkelch13 expression is inversely correlated with PI(3)P levels, the phenotype observed with the *pfkelch13*-upregulated strain presents an intriguing possibility that PI(3)P may be connected to parasite fitness under heat stress (*Mbengue et al., 2015*). Together, these data hint that PI(3)P may play a cytoprotective role in stress responses and facilitate parasite survival under febrile temperatures.

PI(3)P is a multifunctional lipid regulator that controls vesicular trafficking, protein sorting and autophagy in many model organisms (*Mayinger, 2012*; *Balla, 2013*). The *Plasmodium* genome encodes a single PI3K that primarily synthesizes PI(3)P and is essential for intraerythrocytic parasite growth (*Tawk et al., 2010*; *Vaid et al., 2010*; *Zhang et al., 2018*). During the intraerythrocytic cycle, *P. falciparum* generates more PI(3)P at the trophozoite and schizont stages where this lipid localizes to the apicoplast and the digestive vacuole (DV) (*Tawk et al., 2010*). Notably, the DV is an acidic organelle where hemoglobin degradation and heme detoxification occur, and may serve as an acute sensor for cellular stresses similar to its functional counterpart: the lysosome (*Goldberg, 2013*). In mammalian cells, lysosomes may undergo membrane destabilization in response to different stresses, which can lead to programmed cell death or necrosis (*Olson and Joyce, 2015*; *Kirkegaard and Jäättelä, 2009*; *Li and Kane, 2009*). Although the molecular function of PI(3)P in hemoglobin trafficking to the DV has been reported (*Vaid et al., 2010*), its potential role in modulating *Plasmodium* stress responses has not yet been investigated. Here, we present a novel function of PI(3)P in stabilizing the *Plasmodium* DV at clinically relevant febrile temperatures. Through integrating chemical, biochemical and conditional genetic approaches, we identified PfHsp70-1 to be a PI(3)P effector protein that facilitates DV integrity and contributes to parasite fitness during heat shock.

## Results

### Targeting PI(3)P synthesis reduces *Plasmodium* parasite fitness under heat stress

*P. falciparum* parasites are commonly exposed to heat stress in acute malaria patients from the early ring stage to the onset of schizogony. To interrogate if PI(3)P facilitates the survival of mature parasites under heat shock, we first tested to what extent these parasites can tolerate heat. Synchronized

*P. falciparum* 3D7 was cultured at 28–37 hr post-invasion (hpi) at a physiologically relevant febrile temperature (40°C) for 3–12 hr, followed by recovery at 37°C for 45 hr (*Figure 1—figure supplement 1A*). We observed growth inhibition that generally correlated with the duration of the heat shock and no developmental arrest, as >97% of the parasites were trophozoites even after a 12 hr heat shock (*Figure 1—figure supplement 1*). Prolonged heat shock ($\geq$9 hr) led to a 32–41% reduction in parasite loads (p<0.05, unpaired t-test) compared to the non-heat-shocked control (0 hr HS). However, heat shock for up to 6 hr resulted in no significant reduction in parasite growth/survival (3 hr HS: p=0.27, 6 hr HS: p=0.12), indicating an intrinsic capacity of mature stage parasites to tolerate heat stress (*Figure 1—figure supplement 1C*). Thus, we used 6 hr heat shock for most of the following experiments since the parasites can tolerate this treatment without death in our assays.

Two compounds with different chemical scaffolds, Wortmannin and LY294002, have been shown to reduce PI(3)P levels in *P. falciparum* and *Toxoplasma gondii* (a closely related parasite) (*Mbengue et al., 2015*; *Tawk et al., 2010*; *Bansal et al., 2017*; *Dalal and Klemba, 2015*; *Tawk et al., 2011*; *Kitamura et al., 2012*; *Besteiro et al., 2011*; *Stutz et al., 2012*). Here, we utilized these compounds to probe PI(3)P function and necessity for *P. falciparum* survival under heat stress. Trophozoite-stage parasites (32 hpi) were treated with the inhibitors (0.3–1.5-fold $EC_{50}$ concentrations) and subjected to a 6 hr heat shock. After treatment, parasites were returned to 37°C and the relative parasite loads were measured 34 hr after reinvasion. Upon heat shock, Wortmannin and LY294002 decreased parasite loads by 48 ± 32% and 18 ± 10%, respectively, compared to non-heat shock treatment (*Figure 1A*). A similar phenomenon was observed with drug treatment of schizonts (38 hpi), where a 39 ± 19% and 37 ± 3.2% parasite reduction was detected after heat shock of Wortmannin- and LY294002-treated parasites, respectively (*Figure 1—figure supplement 2*). No inhibition was detected in the negative control DMSO after heat shock, indicating an increased sensitivity to heat in PI(3)P-deficient parasites. To further demonstrate that PI(3)P deficiency renders parasites heat intolerant, we performed a standard dose response study with 6 hr heat exposure at the trophozoite stage. We detected a two-fold decrease in the LY294002 $EC_{50}$ value after heat shock (*Figure 1B*). To control for possible off-target effects of the compound, we also tested an inactive LY294002 analog, LY303511, which does not disrupt *P. falciparum* PI(3)P levels (*Mbengue et al., 2015*). LY303511 inhibited *P. falciparum,* but this inhibition was not heat-dependent (*Figure 1C*). Wortmannin was similarly tested in our assay, but an $EC_{50}$ shift was not observed using our standard assay format (*Figure 1—figure supplement 3B*), possibly due to compound instability in complete medium (*Tawk et al., 2010*). Indeed, using another format with heat exposure immediately after Wortmannin addition resulted in a ~ 1.3 fold decrease in the $EC_{50}$ value (*Figure 1D*). In contrast, a range of antimalarial drugs including atovaquone (electron transport inhibitor), pyrimethamine (folate synthesis inhibitor), quinacrine (unresolved mechanism) and lapachol (unresolved mechanism) did not exhibit a drug sensitivity change with heat exposure (*Figure 1—figure supplement 3*). These compounds were selected due to their established inhibition of *Plasmodium* and their diverse modes of action. These results suggest a distinct cytoprotective role for PI(3)P during malaria febrile episodes.

We then investigated if *Plasmodium* parasites actively generate more PI(3)P in response to heat stress. Total parasite lipids were extracted with and without heat shock, spotted onto nitrocellulose membranes and probed with a PI(3)P-specific binding peptide, 2xFyve (*Gillooly, 2000*), to quantify relative PI(3)P levels. Intriguingly, PI(3)P levels in *P. falciparum* parasites increased by 1.3–2.3 fold with heat shock, while uninfected RBCs exhibited no change after heat shock (*Figure 1E*). Our data indicates heat-shock-induced PI(3)P accumulation occurs in the mature stage parasites, which could contribute to parasite survival during malarial fever.

## Inhibiting PI(3)P synthesis disrupts subcellular localization of a digestive vacuolar protein during heat stress

Digestive organelles such as lysosomes are known to function as an early sensor for different stress stimuli in eukaryotes. These stress stimuli can cause lysosomal membrane permeabilization, which in turn leads to cell death (*Olson and Joyce, 2015*; *Kirkegaard and Jäättelä, 2009*; *Ingemann and Kirkegaard, 2014*; *Boya and Kroemer, 2008*). Since PI(3)P is localized to the *Plasmodium* digestive vacuole (DV) (*Tawk et al., 2010*), we interrogated if PI(3)P prevents DV membrane permeabilization upon heat shock. We first employed a transgenic parasite expressing a GFP-tagged, DV-resident protein, plasmepsin II (PM2GT) (*Klemba et al., 2004*) to probe the effect of heat shock on its

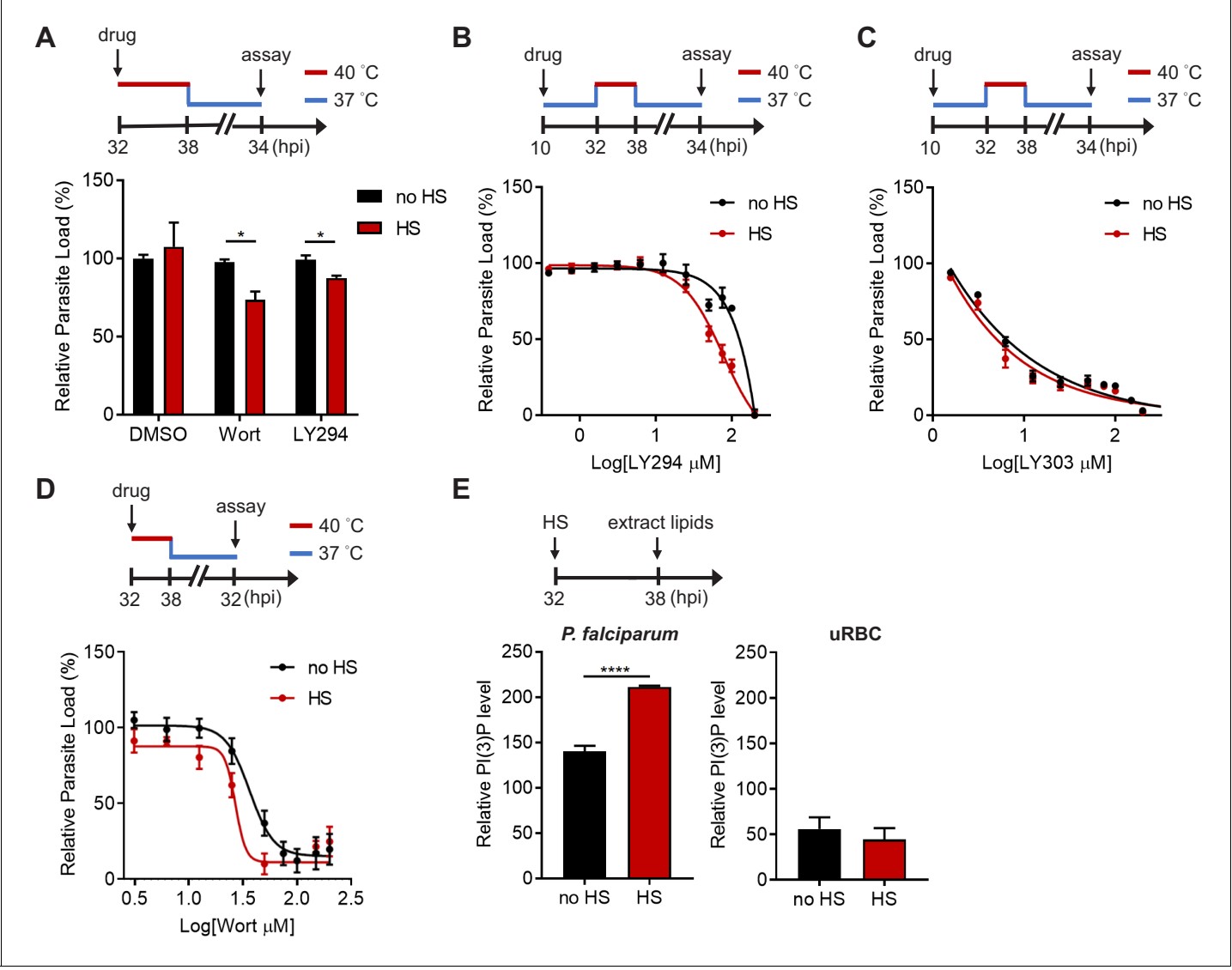

**Figure 1.** PI(3)P reduction sensitizes heat-shock-induced *Plasmodium* parasite death. (**A**) Inhibition of PI(3)P synthesis reduces parasite fitness under heat shock. Assay schematic above plot shows drug administration 32 hpi and analysis at 34 hr after reinvasion (50 hr drug treatment) for the growth assay. *P. falciparum* 3D7 parasites were subjected to a 6 hr heat shock (32–38 hpi) (red line), followed by recovery at 37°C (blue line). Parasite loads were normalized to the DMSO-treated, non-heat-shocked control. Parasites that received (red bar) or did not receive (black bar) a 6 hr heat shock in the presence of 0.1% DMSO, 20 μM Wortmannin (Wort) or 40 μM LY294002 (LY294) are shown. Representative data of three biological replicates is shown (n = 3). *p<0.05 (unpaired t-test). (**B–D**) Dose response curves for LY294002 (**B**), LY303511 (LY303, inactive analog of LY294002) (**C**) and Wortmannin (**D**) inhibition of *P. falciparum* 3D7 with (red circles) or without (black circles) heat shock (HS). Assay schematics shown above plots indicate times of drug addition and assay analysis after reinvasion. Parasites received a 6 hr heat shock (32–38 hpi) (red line) and were maintained at 37°C (blue line) before and after heat shock. Representative data of two biological replicates are shown (n = 3). (**E**) The heat shock effect on intra-parasitic levels of PI(3)P. Assay schematic above plot indicates heat shock treatment and cell harvesting for lipid extraction. Total lipids of *P. falciparum* (left panel) and uninfected red blood cells (uRBC, right panel) that received (red bar) or did not receive (black bar) a 6 hr heat shock were extracted and spotted on a nitrocellulose membrane. The relative amounts of PI(3)P were quantified using a PI(3)P-specific binding peptide, 2xFyve. Representative data of three biological replicates is shown (n = 3). ****p<0.0001 (unpaired t-test). The bars represent mean ± SEM.

The online version of this article includes the following figure supplement(s) for figure 1:

**Figure supplement 1.** *Plasmodium* parasites at the trophozoite and early schizont stages tolerate a 6 hr heat shock.
**Figure supplement 2.** Inhibiting PI(3)P biogenesis reduces *P. falciparum* schizont growth during heat shock.
**Figure supplement 3.** The heat-shock-induced drug hypersensitivity is not detected with other antimalarial drugs.

localization. The mean fluorescence intensities within the DV (DV MFIs) and relative intra-parasitic distribution of PM2GT (MFI ratio (DV/non-DV)) were quantitatively measured to detect abnormalities in PM2GT localization. No aberrant DV morphology or PM2GT mislocalization was found after a 6 hr heat shock, demonstrating the intrinsic stability of the DV during heat shock (p=0.64; *Figure 2*, DMSO control).

Next, we examined if chemically inhibiting PI(3)P production affects DV membrane stability and morphology. However, Wortmannin treatment resulted in a high background signal and PM2GT mislocalization even at 37°C (no heat shock), making it difficult to identify heat-dependent changes (*Figure 2B,D*). In contrast, LY294002-treated parasites showed normal DV morphology and PM2GT localization at 37°C, with DV MFIs comparable to the DMSO control (*Figure 2B–D*). Interestingly, PM2GT redistributed out of the DV after heat shock, concomitant with a reduction in DV-localized GFP signal (*Figure 2B–D*). These changes were not observed with the inactive LY294002 analog (LY303511).

However, the temperature-dependent plasmepsin II redistribution and reduced DV signal in the PM2GT strain could also be due to defective protein trafficking or decreased protein translation. Therefore, we used LysoTracker Red, which normally accumulates in the *Plasmodium* DV, to probe DV membrane integrity that is independent of protein synthesis and trafficking (*Tomlins et al., 2013*). To complete this study, we first optimized our live cell confocal microscopy assay conditions to avoid laser-induced DV photolysis (*Wissing et al., 2002*; *Rohrbach et al., 2005*). Then we measured the accumulated LysoTracker Red signal to compare DV membrane integrity between heat-shocked and non-heat-shocked parasites. In agreement with our PM2GT experiment, 6 hr heat shock did not affect DV integrity (p=0.12; *Figure 3A–C*, DMSO control). However, both Wortmannin and LY294002 destabilized the DV after heat shock (*Figure 3B,C*). In contrast, LY303511-treated parasites showed DV signals comparable to the DMSO control with or without heat shock.

We next tested different antimalarial drugs using the same experimental setup to determine if DV membrane destabilization was triggered by PI(3)P deficiency or a non-specific stress response associated with drug exposure. Three representative compounds (lapachol, atovaquone and pyrimethamine) with different modes of action and potencies ranging from 1 nM to 46 µM had no effect on DV stability at 37°C and 40°C (*Figure 3—figure supplement 1A–E*). Notably, the reported mechanisms of action of these compounds are distinct from that of Wortmannin and LY294002, and thus likely do not affect *Plasmodium* PI(3)P levels (*Gregson and Plowe, 2005*; *Sibley et al., 2001*; *Birth et al., 2014*; *Blasco et al., 2017*). We also tested an artemisinin-related compound, artesunate, and found that it induced heat shock-dependent DV destabilization (*Figure 3D,E*). Intriguingly, like Wortmannin and LY294002, artemisinin and its analogs have been reported to disrupt PI(3)P levels in *P. falciparum* parasites (*Tilley et al., 2016*; *Mbengue et al., 2015*). Based on our microscopy data, there is no correlation between DV MFI reduction and the fold $EC_{50}$ concentrations (antimalarial potency) applied in this study (p=0.34, Pearson's correlation coefficient r = −0.47; *Figure 3—figure supplement 1*), consistent with the proposal that DV permeabilization is triggered by reduced PI(3)P levels and not a general stress response. Altogether, these data suggest a positive role for PI(3)P in maintaining DV membrane integrity during heat shock.

## Heat-induced digestive vacuole destabilization in PI(3)P-deficient parasites is not a general consequence of cell death and is reversible

To study if DV membrane permeabilization upon heat shock is an early cellular event associated with death, we measured mitochondrial accumulation of JC-1 as a proxy of parasite survival (*Paloque et al., 2016*; *Peatey et al., 2015*; *Lee et al., 2014*; *Rathore et al., 2011*). JC-1 produces green fluorescence in the cytosol, while mitochondrial accumulation is associated with an increase in red fluorescence, which requires maintenance of the mitochondrial membrane potential. Cell death in *Plasmodium* parasites is characterized by mitochondrial depolarization, which can be detected as a decrease in the red-to-green fluorescence intensity ratio (*Lee et al., 2014*; *Rathore et al., 2011*). We first determined the basal mitochondrial membrane potential by measuring the JC-1 red-to-green fluorescence ratio in live parasites (*Figure 4*, DMSO negative control). A mitochondrial depolarizing agent, CCCP, was also applied as a positive control (*Figure 4*, CCCP positive control). Our data show that mitochondria remained polarized after LY294002 and heat treatments, as revealed by their JC-1 red/green ratios compared to the DMSO controls (*Figure 4*). These results are consistent with LY294002- and heat-shock-induced DV membrane destabilization occurring prior to

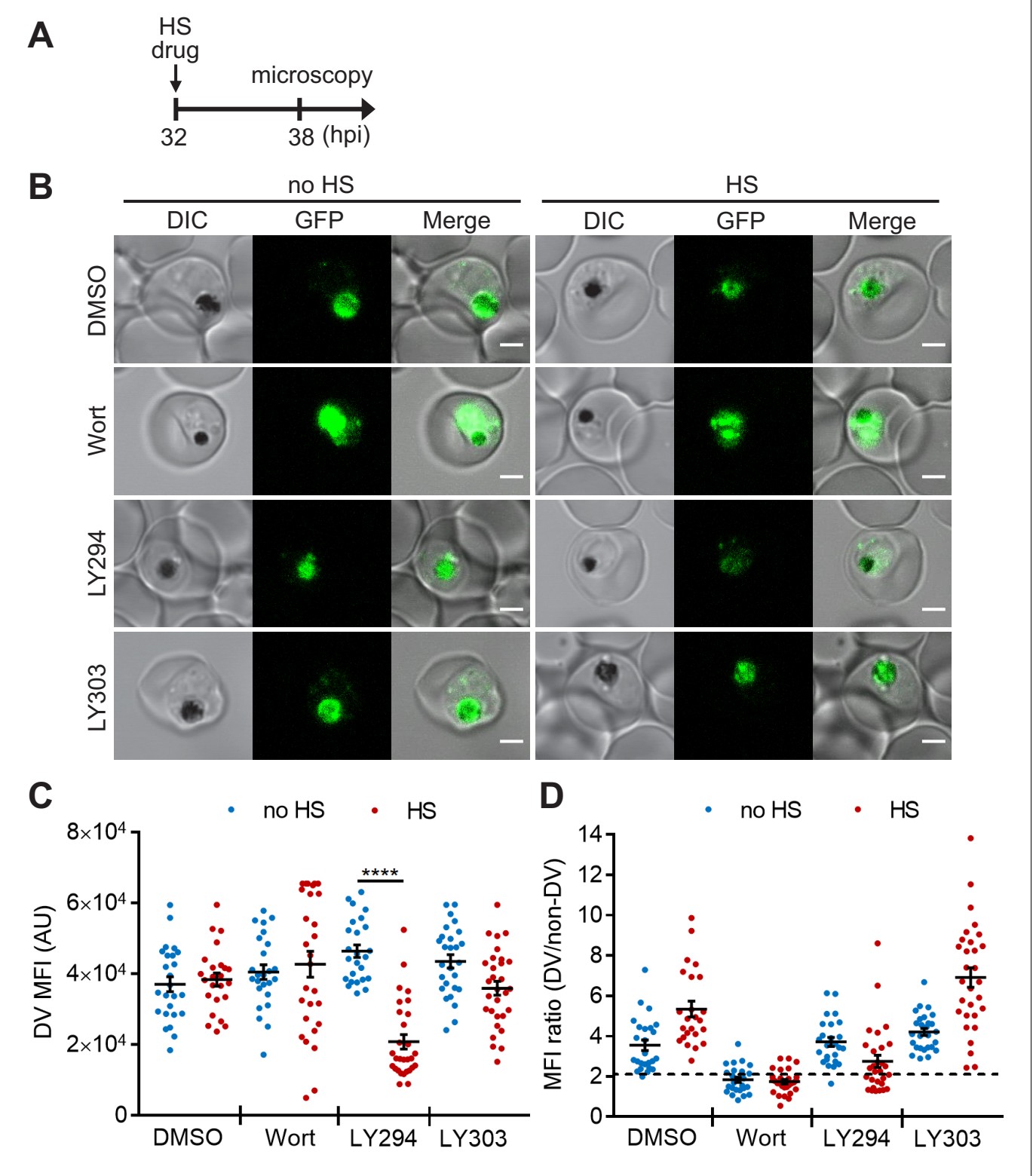

**Figure 2.** Targeting PI(3)P synthesis disrupts the subcellular localization of a DV-resident protein plasmepsin II under heat stress. *P. falciparum* PM2GT parasites were treated with 0.1% DMSO, 20 µM Wortmannin (Wort), 40 µM LY294002 (LY294) or 40 µM LY303511 (LY303) at 37°C (no HS) or 40°C (HS) for 6 hr. (**A**) Assay schematic showing drug and heat shock treatment at 32 hpi, followed by microscopy at 38 hpi. (**B**) Representative images from live cell confocal microscopy are shown. GFP, green fluorescent protein-tagged plasmepsin II; DIC, differential interference contrast. Scale bar, 2 µm. (**C**) Mean fluorescence intensities of DVs (DV MFIs) in heat-shocked (red) and non-heat-shocked (blue) parasites were quantified. Representative data of three

*Figure 2 continued on next page*

*Figure 2 continued*

biological replicates is shown (n > 20). ****p<0.0001 (Welch's t-test). The bars represent mean ± SEM. (D) The ratios of mean fluorescence intensities within DVs (DV MFI) to that in the non-DV areas (non-DV MFI) in heat-shocked (red) and non-heat-shocked (blue) parasites. The MFI ratio <2 (dotted line) indicates mislocalization of plasmepsin II-GFP to the non-DV regions. Representative data of three biological replicates is shown (n > 20).

The online version of this article includes the following source data for figure 2:

**Source data 1.** The DV MFIs and MFI ratios in *Figure 2*.

mitochondrial membrane potential loss. In contrast, mitochondrial membrane depolarization was observed in ~55% of the Wortmannin-treated parasites after heat shock (*Figure 4*). From Lyso-Tracker loading experiments, >70% of the parasites showed DV membrane destabilization following the same treatment, which suggests that a sub-population of parasites may undergo DV

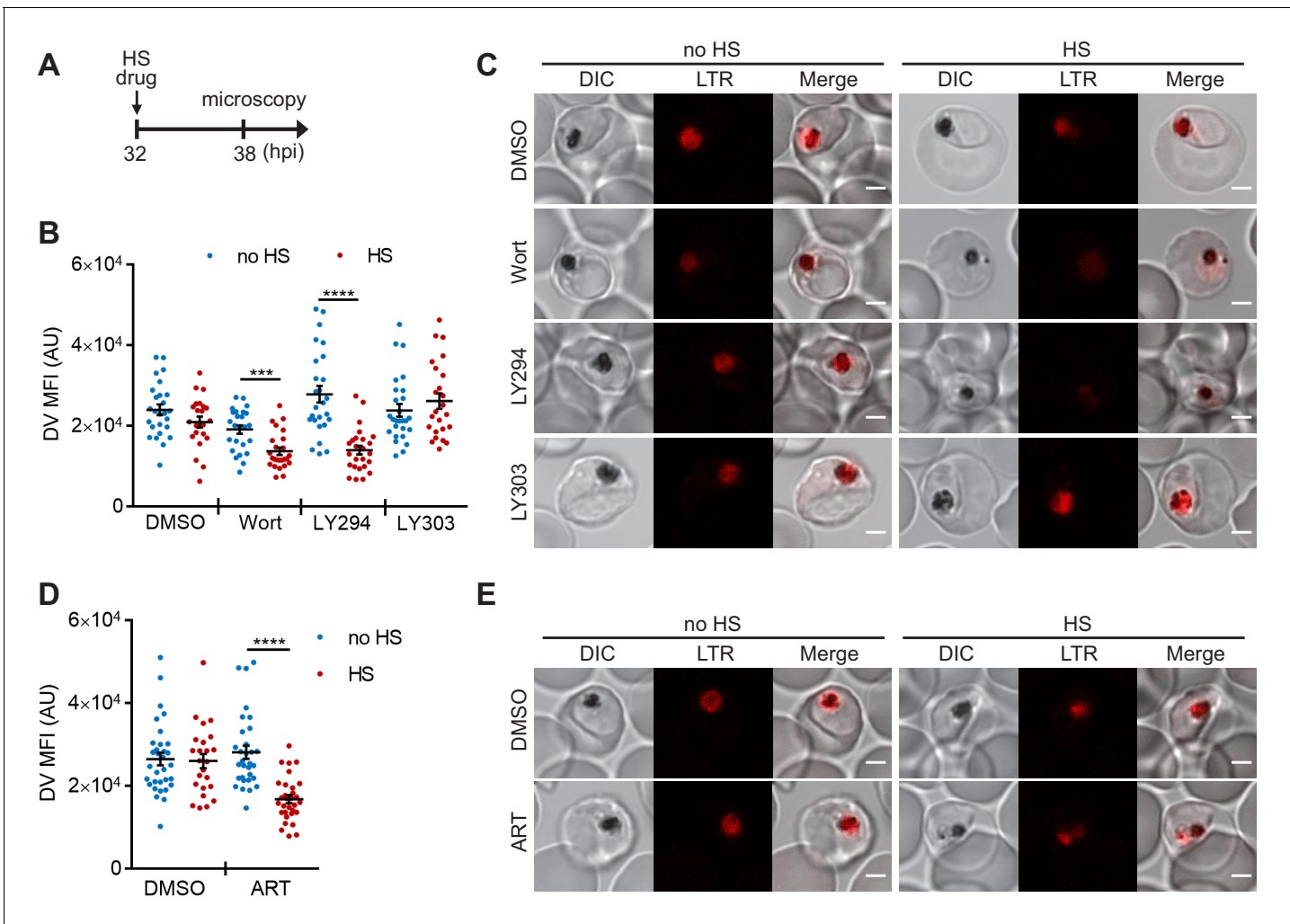

**Figure 3.** PI(3)P biogenesis maintains *Plasmodium* DV integrity under heat shock. *Plasmodium* 3D7 parasites were loaded with LysoTracker Red to stain acidic organelles and treated with 0.1% DMSO, 20 µM Wortmannin (Wort), 40 µM LY294002 (LY294), 40 µM LY303511 (LY303) or 20 nM artesunate (ART) at 37°C (no HS) or 40°C (HS) for 6 hr. (A) Assay schematic showing drug and heat-shock treatment at 32 hpi, followed by live cell confocal microscopy at 38 hpi. (B and D) Mean fluorescence intensities of DVs (DV MFIs) in heat-shocked (red) and non-heat-shocked (blue) parasites were quantified. Representative data of three biological replicates are shown (n > 20). ***p<0.001; ****p<0.0001 (Welch's t-test). The bars represent mean ± SEM. (C and E) Representative images from live cell confocal microscopy are shown. LTR, LysoTracker Red; DIC, differential interference contrast. Scale bar, 2 µm.

The online version of this article includes the following source data and figure supplement(s) for figure 3:

**Source data 1.** The DV MFIs in *Figure 3*.

**Figure supplement 1.** The heat-shock-induced DV destabilization does not correlate with antimalarial potency of the inhibitors.

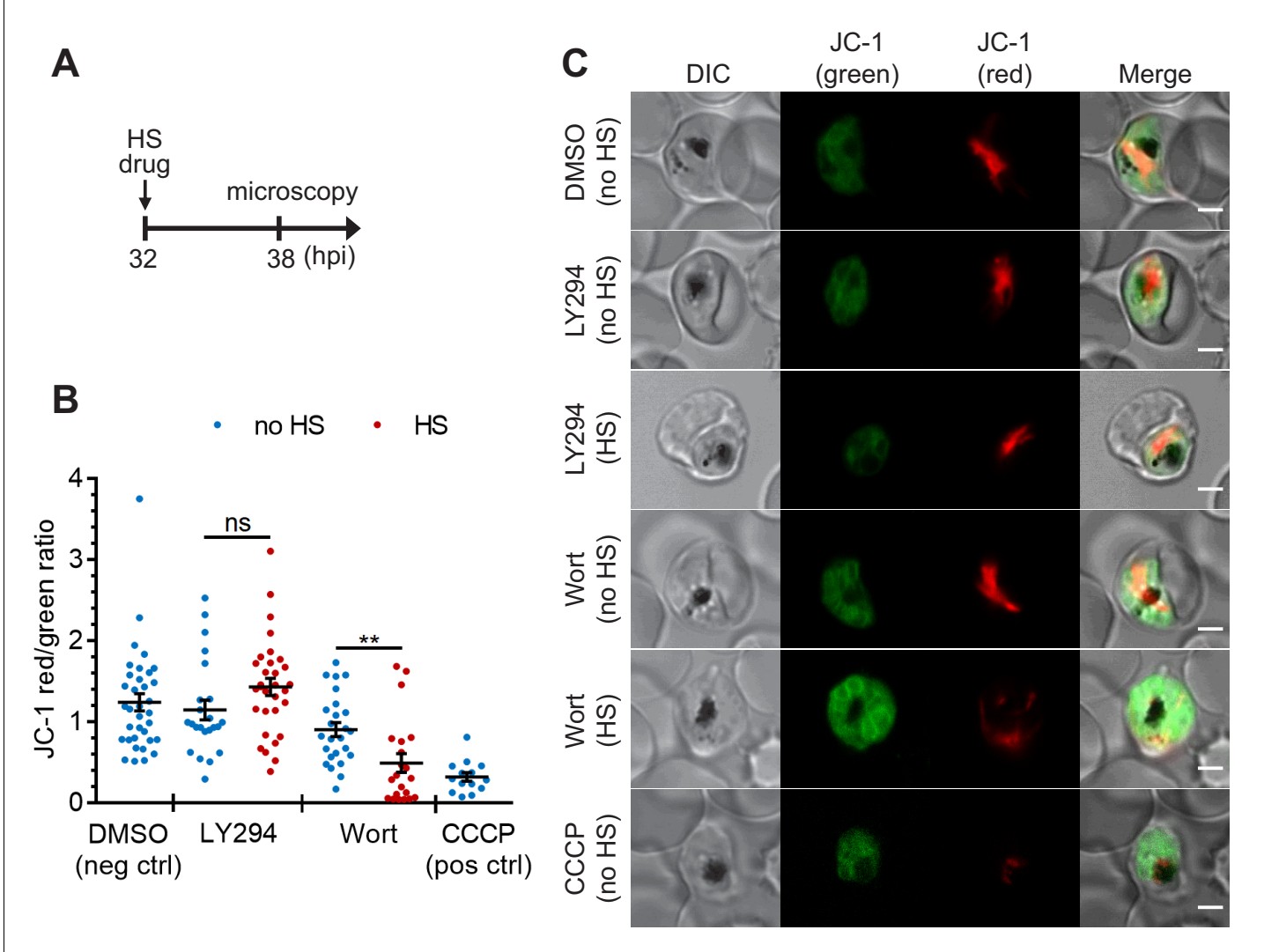

**Figure 4.** The heat-shock-induced DV destabilization is organelle specific and not a result of parasite death in PI(3)P-deficient cells. *Plasmodium* 3D7 parasites were treated with 20 μM Wortmannin (Wort) or 40 μM LY294002 (LY294) at 37°C (no HS) or 40°C (HS) for 6 hr and loaded with JC-1 dye to monitor mitochondrial membrane potential as a marker for parasite viability. DMSO and CCCP were used as negative and positive controls for mitochondrial depolarization, respectively. (**A**) Assay schematic showing drug and heat-shock treatment at 32 hpi, followed by JC-1 loading for microscopy. (**B**) The ratios of JC-1 red fluorescence to JC-1 green fluorescence in heat-shocked (red) and non-heat-shocked (blue) parasites were quantified to determine the degree of mitochondrial membrane depolarization. Representative data of three biological replicates is shown (n > 20). **p<0.01; ns, not significant (Welch's t-test). The bars represent mean ± SEM. (**C**) Representative images from live cell confocal microscopy are shown. JC-1 (green) indicates the monomeric dye in the cytoplasm, while the aggregated JC-1 in the mitochondria of viable parasites emits red fluorescence. DIC, differential interference contrast. Scale bar, 2 μm.

The online version of this article includes the following source data for figure 4:

**Source data 1.** The JC-1 red/green ratios in *Figure 4*.

destabilization prior to mitochondrial depolarization. The higher potency of Wortmannin relative to LY294002 in inhibiting PI3K activity may be a factor contributing to a more mixed phenotype (*Vaid et al., 2010*).

DV permeabilization is generally considered a lethal event since it was assumed that the release of proteases and accompanying cytosolic acidification induces cell death (*Porter et al., 2008*; *Ch'ng et al., 2011*). However, whether this cellular event is a point of no return in *Plasmodium* cell death was unknown. As LY294002 induced heat-dependent DV destabilization without compromising mitochondrial membrane potential, we hypothesized that the DV can return to its normal state

as parasites recover from heat shock. To test this, heat-shocked, LY294002-treated parasites were returned to normal growth conditions (37°C) for 3 hr with or without LY294002 wash out and analyzed by microscopy. We observed LysoTracker Red accumulation in the DV of parasites recovering from heat shock in the absence of LY294002, but not those maintained in the presence of LY294002 (*Figure 5*). This is consistent with the restoration of DV membrane integrity when PI3K inhibition is relieved, and suggests that the DV membrane has a level of resilience to heat stress.

## Identification of PI(3)P-binding proteins in *P. falciparum*

To uncover the underlying mechanism of how PI(3)P stabilizes the DV, we utilized a proteomic method to identify PI(3)P-binding proteins in *Plasmodium* parasites. Fractionation of the intracellular *P. falciparum* erythrocytic parasites can be achieved with saponin, which selectively permeabilizes the host erythrocyte membrane while maintaining the integrity of the parasite plasma membrane. After removal of host proteins, the enriched parasite lysate was incubated with PI(3)P-conjugated beads to pull down *Plasmodium* PI(3)P-binding proteins for MALDI–TOF mass spectrometry analysis. We identified 12 *Plasmodium* proteins that co-precipitated with PI(3)P-conjugated beads (*Table 1*), among which three DV-associated proteins (PfRan, PfAlba1 and PfHsp70-1) (*Lamarque et al., 2008*) were selected for validation studies. Each candidate gene was cloned with a C-terminal poly-His-tag, expressed in yeast, and purified to >95% homogeneity (*Figure 6—figure supplement 1*). The known PI(3)P-specific binding peptide, 2xFyve (*Gillooly, 2000*), was also purified in parallel for an

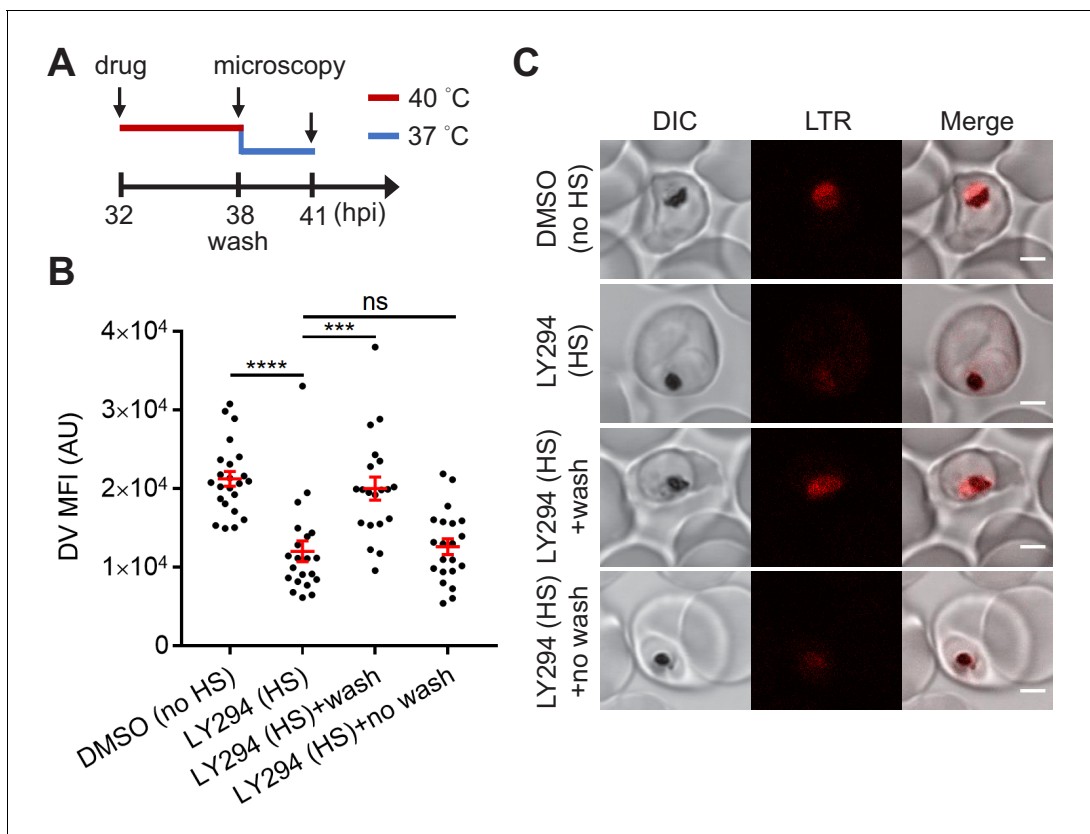

**Figure 5.** Reduced PI(3)P production causes delayed DV recovery. *Plasmodium* 3D7 parasites were loaded with LysoTracker Red (LTR) to stain acidic organelles and treated with 0.1% DMSO or 40 μM LY294002 (LY294) at 37°C (no HS) or 40°C (HS) for 6 hr. (**A**) Assay schematic showing drug and heat treatment (red line) at 32 hpi, followed by recovery at 37°C (blue line) at 38 hpi for 3 hr in the presence (no wash) or absence (wash) of the inhibitor. Confocal microscopy was performed immediately after heat shock (38 hpi) or after recovery (41 hpi). (**B**) Mean fluorescence intensities of parasite DVs (DV MFIs) were quantified. Representative data of three biological replicates is shown (n > 20). ***p<0.001; ****p<0.0001; ns, not significant (Welch's t-test). The bars represent mean ± SEM. (**C**) Representative confocal images are shown. DIC, differential interference contrast. Scale bar, 2 μm.

The online version of this article includes the following source data for figure 5:

**Source data 1.** The DV MFIs in *Figure 5*.

**Table 1.** Identified *P. falciparum* proteins that interacted with PI(3)P.

| Protein name | UniProt accession | Peptide matches | %Coverage* | Apicoplast† | Secretory‡ | Digestive vacuole§ |
|---|---|---|---|---|---|---|
| PfRan | W7JA41 | 6 | 33.2 | - | - | + |
| PfRps4 | W7J × 33 | 4 | 20.5 | - | - | - |
| PfRps19 | W7JZJ7 | 3 | 18.2 | - | - | - |
| PfRps18 | W7JTH8 | 3 | 23.7 | - | + | - |
| PfAlba1 | W7JW62 | 3 | 10.5 | - | - | + |
| PfRps9 | W7K9C7 | 2 | 12.2 | ++ | - | - |
| PF14_0141 | W7JMY4 | 1 | 16 | - | - | - |
| PfHsp70-1 | W7K6C4 | 1 | 1.6 | - | - | + |
| PfRpl3 | W7K5U2 | 1 | 2.3 | - | + | - |
| PFF0885w | W7K862 | 1 | 9.5 | ++ | - | - |
| PF07_0088 | W7K740 | 1 | 5.1 | ++ | - | - |
| MAL7P1.201 | W7KIX5 | 1 | 0.7 | 0 | - | - |

*The coverage of proteins by identified peptides.

†Prediction by the PlasmoAP algorithm based on apicoplast-targeting peptides (++ very likely, 0 uncertain, - unlikely).

‡Prediction by PSEApred based on the amino acid composition (+ secretory, - non-secretory).

§Reported *P. falciparum* proteins associated with the digestive vacuole (+ found, - not found).

assay control. The lipid-binding capacities of these proteins were then probed using a dot blot assay. This assay is routinely used to qualitatively evaluate lipid–protein binding interactions (*Dowler et al., 2002*; *Pal et al., 2012*; *Liu et al., 2011*). Our experiments confirmed direct interactions between PI(3)P and PfRan, PfAlba1 and PfHsp70-1 (*Figure 6A*). Importantly, none of these proteins bound to the control lipid phosphatidylinositol (PI) under the assay conditions. Additionally, we tested another *Plasmodium* protein, PfHop, with no predicted PI(3)P binding as a negative control in the assay. No PI(3)P binding was observed with the same concentration of PfHop, suggesting the assay was detecting specific lipid–protein interactions. We further profiled the lipid-binding specificities of our candidate proteins using a commercial lipid strip that contains 15 different phospholipids. Compared with 2xFyve, which binds to PI(3)P with a greater affinity than other lipids, the three candidate proteins preferentially bound $PI(3,5)P_2$ and phosphatidylinositol monophosphates including PI(3)P with different lipid-binding preferences (*Figure 6B*).

## PfHsp70-1 maintains the *Plasmodium* DV stability under heat stress

Among the identified *Plasmodium* PI(3)P-binding proteins, PfHsp70-1 was of particular interest as its mammalian counterparts have been shown to bind several anionic phospholipids, and through such lipid–protein interactions, human Hsp70 can be recruited to lysosomes where it prevents stress-induced membrane permeabilization (*Morozova et al., 2016*; *Kirkegaard et al., 2010*; *McCallister et al., 2016a*). In addition, mouse Hsp70 was recently reported to bind various phosphoinositides including PI(3)P (*McCallister et al., 2016b*). These findings, alongside with our data, support the hypothesis that PI(3)P recruits PfHsp70-1 to the DV where it stabilizes it in response to heat stress. Unfortunately, studying this lipid–protein interaction in the cellular context is challenging given that both *Plasmodium* PI3K and PfHsp70-1 are predicted to be essential and multifunctional (*Tawk et al., 2010*; *Vaid et al., 2010*; *Zhang et al., 2018*). Moreover, PfHsp70-1 is highly abundant and present in the parasite nucleus and cytoplasm (*Cockburn et al., 2011*; *Pesce et al., 2008*; *Kumar et al., 1991*), thus confounding a possible colocalization study. Indeed, episomally expressed PfHsp70-1-mCherry was localized throughout the parasite cytoplasm, and we did not observe a detectable change in PfHsp70-1 surrounding the DV before or after heat shock (*Figure 7—figure supplement 1*).

To test our hypothesis, we then probed the role of PfHsp70-1 with the small molecule inhibitor 15-deoxyspergualin (15-DSG) that selectively binds to PfHsp70-1 over other *Plasmodium* Hsp70 homologs (*Ramya et al., 2007*). 15-DSG is known to disrupt protein trafficking to another important PI(3)P-enriched organelle, the apicoplast, and to inhibit *Plasmodium* parasite loads, presumably via

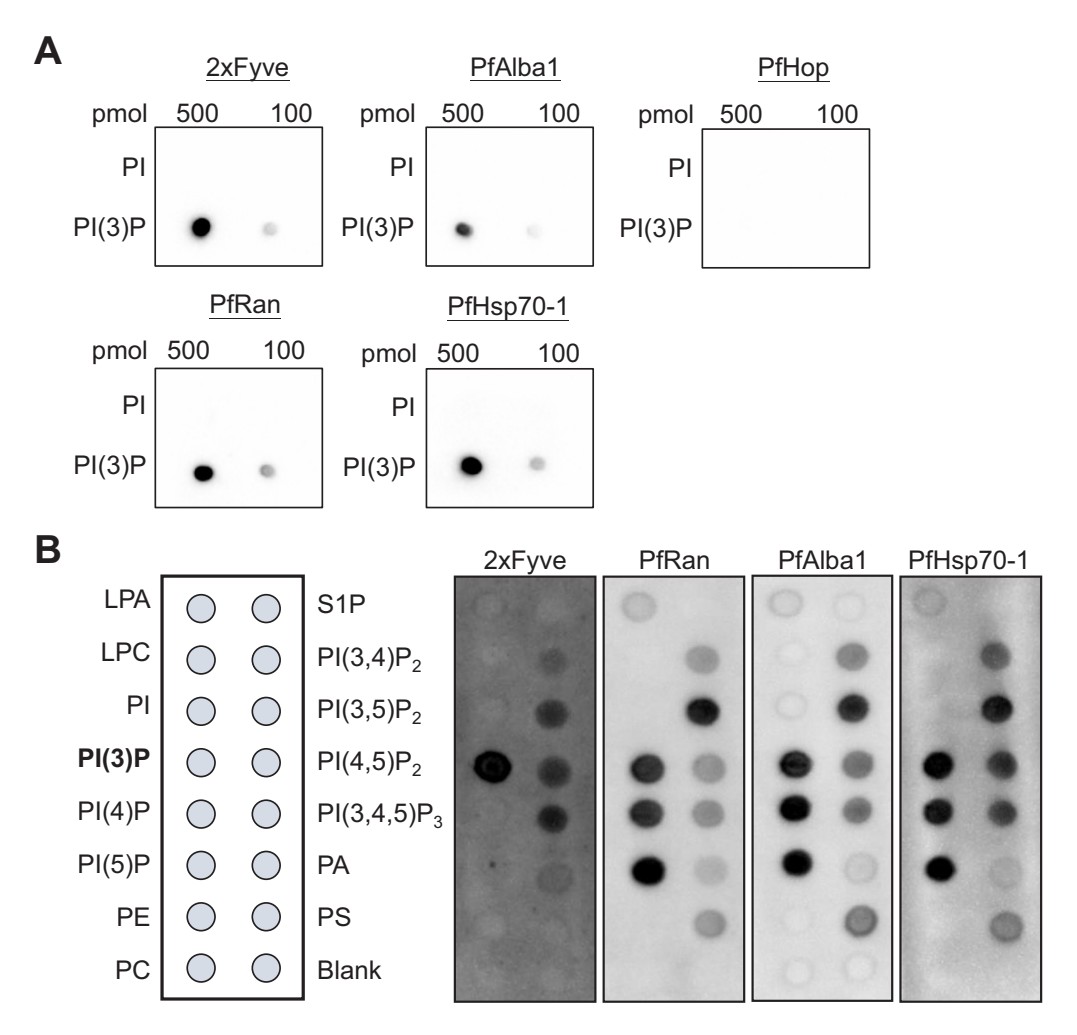

**Figure 6.** Identified *P. falciparum* proteins bind to PI(3)P. (**A**) A proteomic approach identified several *P. falciparum* proteins that may bind to PI(3)P. Candidate proteins (PfRan, PfAlba1 and PfHsp70-1) and a known PI(3)P-specific binding peptide, 2xFyve, were expressed and purified from yeast. Purified His-tagged proteins (>95% pure) were used to probe nitrocellulose membranes spotted with 100 pmol and 500 pmol lipids (PI(3)P and PI). Lipid-binding proteins were detected using electrochemiluminescence. Another His-tagged protein, PfHop, was used as a negative control. Representative data from two–three independent assays is shown. (**B**) Characterization of the lipid-binding specificity for PI(3)P-binding proteins. Map of different lipid spots (100 pmol/spot) indicated (left panel) and specificity of 2xFyve, PfRan, PfAlba1 and PfHsp70-1, respectively, shown. The online version of this article includes the following figure supplement(s) for figure 6:

**Figure supplement 1.** Identified *P. falciparum* PI(3)P-binding proteins can be purified for downstream in vitro binding assays.

targeting PfHsp70-1 (*Tawk et al., 2010*; *Ramya et al., 2007*; *Foth et al., 2003*). Here, we found that parasites treated with 15-DSG exhibited destabilized DVs after 6 hr heat shock, while the same treatment had no effect on the DV when cultured at 37˚C, resembling the phenotype observed in PI(3)P-deficient parasites (*Figure 7A*). To investigate the correlation between reduced DV stability and the PI(3)P-binding capacity of PfHsp70-1, we generated a truncated PfHsp70-1 lacking the C-terminal LID domain. This domain has been previously shown to be the 15-DSG targeting site (*Ramya et al., 2007*). Intriguingly, deletion of the LID domain disrupted the PI(3)P–PfHsp70-1 interaction (*Figure 7—figure supplement 2*), suggesting a mechanistic link between the PI(3)P-binding capacity of PfHsp70-1 and its ability to stabilize the DV under febrile condition.

To more directly study the importance of PfHsp70-1 to DV integrity, we created a transgenic *P. falciparum* parasite that allowed for tunable expression of PfHsp70-1 via an anhydrotetracycline (aTc)-regulated, TetR–aptamer-based system integrated into the 3′ UTR of *pfhsp70-1* in the parasite

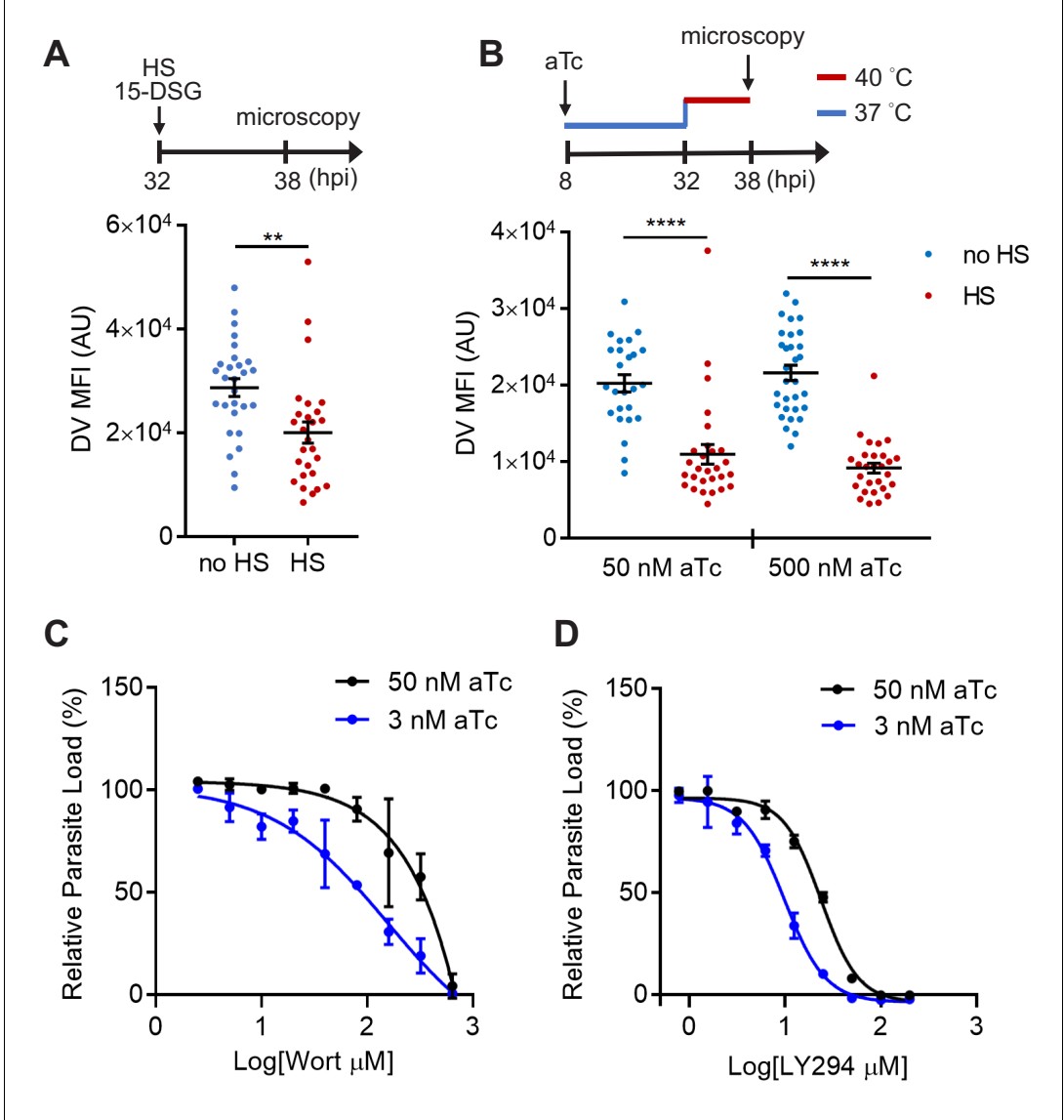

**Figure 7.** PfHsp70-1 depletion causes increased sensitivity to PI3K inhibitors and destabilizes *Plasmodium* DV during heat shock. (**A**) *Plasmodium* 3D7 parasites were loaded with LysoTracker Red and treated with a PfHsp70-1 inhibitor, 15-deoxyspergualin (15-DSG, 1 µM). Assay schematic above plot indicates drug and heat shock treatment at 32 hpi, followed by live cell microscopy at 38 hpi. Mean fluorescence intensities of DVs (DV MFIs) in heat-shocked (HS, red) and non-heat-shocked (no HS, blue) parasites were quantified. Representative data of three biological replicates is shown (n > 20). **p<0.01 (Welch's t-test). (**B**) A tunable PfHsp70-1 parasite line was cultured in 50 nM or 500 nM anhydrotetracycline (aTc) for 24 hr before LysoTracker Red loading at 32 hpi. Assay schematic above plot indicates aTc treatment to modulate PfHsp70-1 expression at 37°C (blue line), followed by a 6 hr heat shock (red line). DV MFIs in heat-shocked (HS, red) and non-heat-shocked (no HS, blue) parasites were quantified. Representative data of two biological replicates is shown (n > 20). ****p<0.0001 (Welch's t-test). (**C and D**) Dose response curves for Wortmannin (Wort) (**C**) and LY294002 (LY294) (**D**) in the PfHsp70-1 knockdown line. Drug sensitivities in the presence of 3 nM (blue circles) and 50 nM (black circles) aTc are shown. Representative data of three–four biological replicates is shown. The bars represent mean ± SEM.

The online version of this article includes the following source data and figure supplement(s) for figure 7:

**Source data 1.** The DV MFIs in *Figure 7*.
**Figure supplement 1.** PfHsp70-1 is localized throughout *P. falciparum* parasites under heat shock and the regular culture conditions.
**Figure supplement 2.** C-terminal LID domain deletion disrupts the PI(3)P-binding capacity of PfHsp70-1.
**Figure supplement 3.** PfHsp70-1 is essential during *P. falciparum* intraerythrocytic cycle.
**Figure supplement 4.** The tunable PfHsp70-1 knockdown strain is sensitive to heat shock.
**Figure supplement 5.** The PfHsp70-1 line has lower PfHsp70-1 expression compared to the wild-type strain 3D7.
**Figure supplement 6.** Inhibiting PI(3)P synthesis sensitizes heat-shock-induced cell death in the PfHsp70-1 line.
**Figure supplement 7.** The heat-shock-induced DV destabilization in the PfHsp70-1 knockdown line is not caused by parasite death.

*Figure 7 continued on next page*

*Figure 7 continued*

**Figure supplement 8.** The hypersensitivity to PI3K inhibitors in PfHsp70-1 knockdown parasites is not a general non-specific phenotype.

chromosome using CRISPR-Cas9 (*Ganesan et al., 2016*). In the presence of high aTc (50 nM), PfHsp70-1 expression is maintained and parasites replicate within erythrocytes (*Figure 7—figure supplement 3*). Upon aTc removal, PfHsp70-1 is depleted and parasite growth is substantially inhibited (*Figure 7—figure supplement 3*). These data indicate that PfHsp70-1 is essential for intraerythrocytic *P. falciparum* replication. A dose response analysis showed that 4.6 nM aTc resulted in 50% reduction of parasite growth, while no effect on parasite growth was observed above 12.5 nM (*Figure 7—figure supplement 4B*). However, parasite viability in this transgenic parasite (maintained in 500 nM aTc) was highly sensitive to heat shock even with up to 1 µM aTc (*Figure 7—figure supplement 4B*). In contrast, removal of aTc did not affect parasite loads at 37°C and 40°C in a control parasite line in which yellow fluorescent protein (YFP) expression is regulated by the same aptamer-based system (*Figure 7—figure supplement 4C*). To further understand the heat-shock sensitivity observed in the PfHsp70-1 line, we examined the relative PfHsp70-1 protein levels with and without heat shock in comparison to the wild-type *P. falciparum* 3D7 line using western blot analysis. Our data show that PfHsp70-1 expression was moderately attenuated in the PfHsp70-1 line both with and without heat shock when compared to the wild-type parasite line (as normally cultured in the complete medium containing 500 nM aTc) (*Figure 7—figure supplement 5*). Thus, the increased heat sensitivity compared to the wild-type and YFP strains was most likely due to a decrease in the basal expression level of PfHsp70-1. Furthermore, treatment with PI3K inhibitors, but not control antimalarial compounds, sensitized the PfHsp70-1 line to heat shock, consistent with our findings in the wild-type 3D7 parasites (*Figure 7—figure supplement 6*).

Upon heat shock, we observed that PfHsp70-1-deficient parasites (50 nM and 500 nM aTc) exhibited DV destabilization (*Figure 7B*), consistent with our 15-DSG data. Importantly, the same heat treatment did not cause mitochondrial membrane permeabilization in the PfHsp70-1 knockdown strain cultured in 500 nM aTc, and only had a moderate effect when cultured in 50 nM aTc (*Figure 7—figure supplement 7*). These data indicate that the expression level of the PI(3)P-binding protein PfHsp70-1 is connected to the maintenance of DV membrane stability during heat stress. Additionally, reduction of PfHsp70-1 levels in the conditional knockdown line (3 nM vs 50 nM aTc) induced hypersensitivity to both Wortmannin and LY294002 by 1.9–2.3 fold. This change in drug sensitivity was not observed with Bafilomycin A, a known anti-*Plasmodium* agent with a different mode of action (*Figure 7C,D* and *Figure 7—figure supplement 8A*). Bafilomycin A targets V-type H$^+$-ATPase and inhibits *Plasmodium* DV acidification (*Saliba et al., 2003*). Our data showing drug hypersensitivity with PfHsp70-1 knockdown suggests that the protein interacts with PI(3)P instead of a random DV component. Furthermore, this hypersensitivity to Wortmannin and LY294002 was not observed in the control YFP line (*Figure 7—figure supplement 8B and C*). Altogether, these results support the proposal that PfHsp70-1 and PI(3)P act in a coordinated manner to influence *P. falciparum* DV membrane stability during heat stress.

## Discussion

In this study, we identified a novel function for PI(3)P in maintaining *P. falciparum* digestive vacuole (DV) stability under heat-shock conditions simulating malarial fever. Intricate studies including live cell microscopy correlated PI(3)P deficiency with DV permeabilization and subsequent parasite death after heat exposure. The lysosome has long been recognized as an acute sensor for stress stimuli, including oxidative, osmotic, and heat stress, whereby lysosomal membrane destabilization triggers apoptosis or necrosis (*Kirkegaard and Jäättelä, 2009*; *Boya and Kroemer, 2008*; *Wang et al., 2018*). Similarly, *Plasmodium* DV destabilization can result in inefficient hemoglobin digestion, oxidative stress and undesired protein degradation in the parasite cytoplasm, which may eventually lead to impaired growth or cytotoxicity (*Ch'ng et al., 2011*). Yet, a connection between the DV and cellular stress response in *Plasmodium* parasites was unresolved. In human vascular endothelial cells, suppressed PI(3)P synthesis causes the release of the protease cathepsin B from the lysosome and in turn sensitizes cells to lysosomal-dependent cell death (*Kirkegaard and Jäättelä, 2009*;

*Madge et al., 2003*). Additionally, inhibiting PI(3)P biogenesis impedes lysosomal function and leads to endolysosomal membrane damage in neurons (*Miranda et al., 2018*). Thus, we hypothesized that PI(3)P has a conserved functional role in stabilizing the acidic DV in *P. falciparum*. This function is especially intriguing in *Plasmodium*, since the DV is an attractive drug target for malaria treatment (*Olliaro and Goldberg, 1995*). Our findings that PI(3)P contributes to the stability of this specialized organelle hint at a drug combination strategy for increasing parasite susceptibility to DV permeabilization inhibitors.

PI(3)P effector proteins in *Plasmodium* parasites were largely unknown, but we predicted a stress response protein may be associated with PI(3)P to stabilize the DV under heat stress. To facilitate the elucidation of the molecular mechanism of lipid-dependent DV membrane stabilization, a proteomic strategy was employed to identify *P. falciparum* PI(3)P-binding proteins. Three identified PI(3)P-binding proteins were validated using protein–lipid overlay assays with purified, recombinant *P. falciparum* proteins. Among the validated proteins, PfHsp70-1 was particularly interesting as its mammalian homologs prevent photo-induced lysosomal membrane damage by associating with anionic membrane lipids (*Kirkegaard et al., 2010*; *Nylandsted et al., 2004*). We observed PfHsp70-1 binding to PI(3)P and other lipids, but its lower affinity to phosphatidylinositol triphosphate, phosphatidylserine and some phosphatidylinositol bisphosphates indicates that the binding is not due to a nonspecific electrostatic association. While the utilized protein–lipid overlay assay is widely used to identify lipid ligands for a protein, it does not provide quantitative information regarding binding affinity. Future studies employing liposomes with established fluorescence spectroscopy methods (*Lu et al., 2012*; *Saliba et al., 2015*) could quantify binding affinity differences to reveal possible PfHsp70-1 selectivity for different lipids. Notably, PI(5)P and PI(3,5)P$_2$ (that were associated with PfHsp70-1 in vitro) appear to be absent in *Plasmodium* parasites (*Tawk et al., 2010*). Consistently, biochemical studies of mouse Hsp70 demonstrate that it binds to PI(3)P and other lipids (including different phosphatidylinositol monophosphates), supporting a conserved interaction with membranes (*Morozova et al., 2016*; *McCallister et al., 2016a*; *McCallister et al., 2016b*; *McCallister et al., 2015*). In agreement with the cytoprotective role of Hsp70s in stabilizing lysosomal membranes (*Olson and Joyce, 2015*; *Kirkegaard and Jäättelä, 2009*; *Nylandsted et al., 2004*), we demonstrated the importance of PfHsp70-1 in *P. falciparum* DV stability using a genetically modified PfHsp70-1 knockdown strain and a small molecule PfHsp70-1 inhibitor. Importantly, the reduction of PfHsp70-1 levels increased the sensitivity of *P. falciparum* to PI3K inhibitors, suggesting that PfHsp70-1 and PI(3)P may act in a common pathway. Previous studies have also hinted at a link between PI(3)P and PfHsp70-1 in *Plasmodium*. A transcriptomic analysis revealed that heat shock caused a 5.3-fold upregulation of *pfhsp70-1* and a 2.9-fold downregulation of a *P. falciparum* phosphoinositide phosphatase (PF13_0285) (*Oakley et al., 2007*). Additionally, *pfpi3k* and *pfhsp70-1* seem to be co-upregulated in artemisinin-resistant strains, collectively supporting a relationship between PI(3)P and PfHsp70-1 in response to cellular stresses (*Tilley et al., 2016*; *Mbengue et al., 2015*; *Witkowski et al., 2010*).

The mechanism by which PI(3)P and PfHsp70-1 maintain DV stability under stress remains to be resolved, but one possibility could involve the ubiquitin–proteasome pathway (*Figure 8*). In eukaryotes, hyperthermia can induce protein denaturation and aggregation (*Ritchie et al., 1994*; *Roti Roti, 2008*), but the ubiquitin–proteasome machinery helps maintain proteostasis by eliminating unfolded/misfolded cytotoxic proteins (*Aminake et al., 2012*). Likewise, *Plasmodium* has conserved and functional ubiquitin–proteasome machinery with >50% of the proteome harboring at least one ubiquitination site (*Aminake et al., 2012*). In accordance with this hypothesis, artemisinin-resistant parasites usually have an enhanced ubiquitin–proteasome pathway concurrent with elevated PI(3)P levels (*Paloque et al., 2016*). Moreover, Hsp70s can bind to exposed hydrophobic regions on unfolded/misfolded proteins to recruit E3 ubiquitin ligases for proteasomal degradation (*McDonough and Patterson, 2003*; *Esser et al., 2004*). However, our preliminary data showed that inhibiting PI(3)P production did not affect K48-ubiquitination levels in *P. falciparum* under heat shock (*Figure 8—figure supplement 1*). To explore a connection between the proteasome and PI(3)P-dependent phenotype, we utilized the proteasome inhibitor bortezomib as a probe. Bortezomib is a known antimalarial agent that strongly synergizes with artemisinin (*Tilley et al., 2016*; *Aminake et al., 2012*; *Kreidenweiss et al., 2008*; *Reynolds et al., 2007*). Interestingly, inhibiting proteasomal activity caused a reduction in LysoTracker Red accumulation in the DV (*Figure 8—figure supplement 2*), reminiscent of the increased membrane permeabilization observed in PI(3)P-

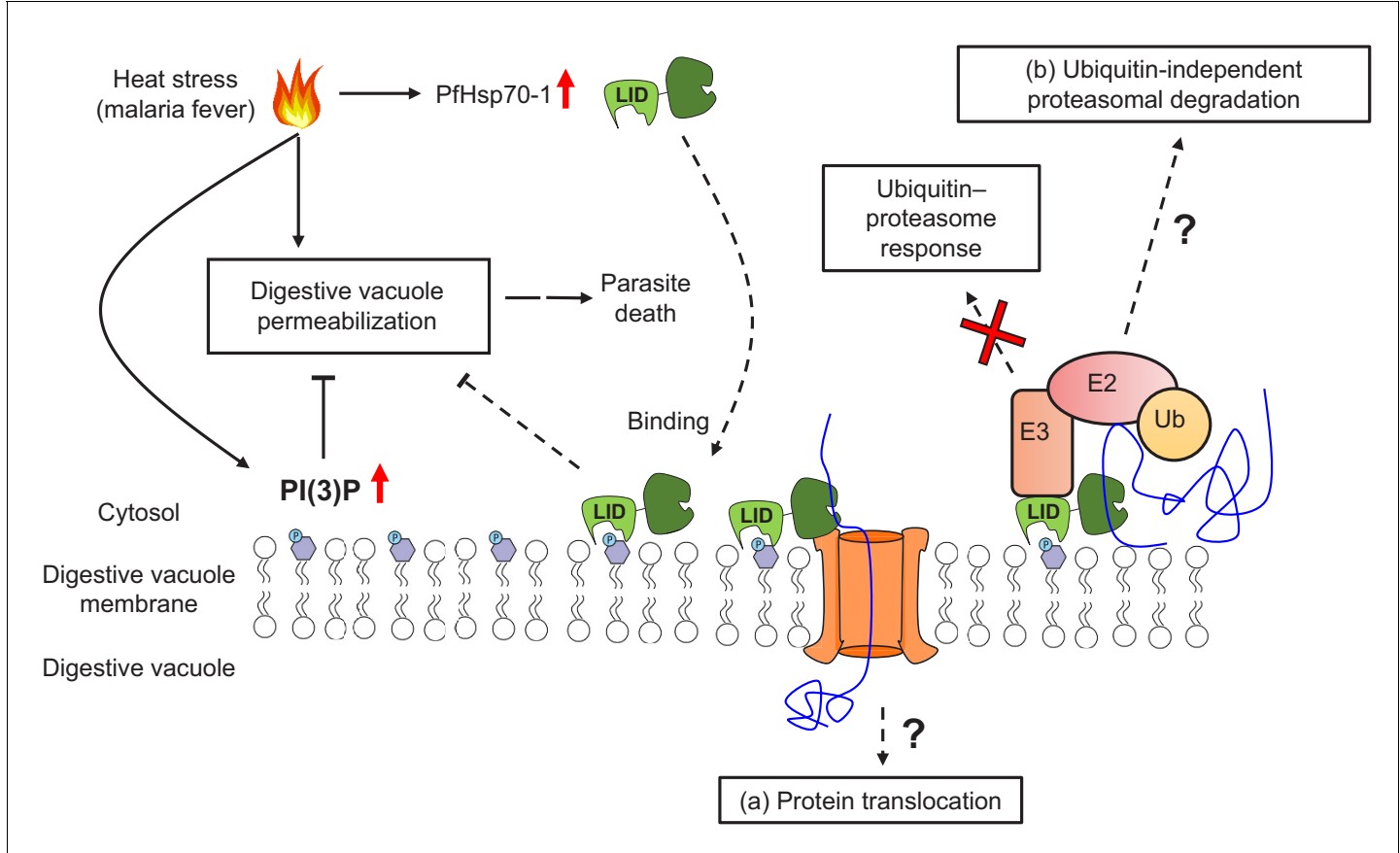

**Figure 8.** A model for PI(3)P-mediated heat stress response in *P. falciparum*. PI(3)P accumulates in *P. falciparum* during febrile episodes and prevents membrane destabilization of the digestive vacuole (DV), thus increasing parasite fitness under heat stress. PfHsp70-1 may be recruited to the DV via PI (3)P binding and contribute to the cytoprotective function. Possible mechanisms of this lipid–protein interaction may include (a) PfHsp70-1-mediated translocation of DV proteins that maintain the membrane integrity from the inner leaflet of the acidic compartment or (b) proteasomal degradation pathway that removes local proteotoxic stress.

The online version of this article includes the following figure supplement(s) for figure 8:

**Figure supplement 1.** Inhibiting PI(3)P production does not alter the overall K48-ubiquitination level in *P. falciparum* under heat shock.
**Figure supplement 2.** Targeting 26S proteasome may sensitize the heat shock-induced DV destabilization.

and PfHsp70-1-depleted parasites under heat shock. But it remains to be determined if PI(3)P and PfHsp70-1 function through the proteasome-mediated unfolded protein response in a ubiquitin-independent manner. Future studies involving the PfHsp70-1 interactome and its possible connection to proteasome-dependent protein degradation may address this question.

Alternatively, PI(3)P and PfHsp70-1 may modulate membrane stability via the protein translocation machinery (*Figure 8*). In other eukaryotes, Hsp70s bind to transit peptides to facilitate protein translocation into different membrane compartments (*Zhang and Glaser, 2002*; *Dudek et al., 2015*). In *P. falciparum*, it has been proposed that PfHsp70-1 is crucial for nuclear-encoded protein trafficking into the apicoplast, another PI(3)P-enriched parasite organelle (*Tawk et al., 2010*). Mutagenesis of putative Hsp70-binding sites within transit peptides, or PfHsp70-1 inhibition with 15-DSG, reduced apicoplast protein translocation in *P. falciparum* (*Ramya et al., 2007*; *Foth et al., 2003*). It is thus conceivable that a PI(3)P–PfHsp70-1 association has a similar function in DV protein trafficking to either remove toxic protein aggregates from the cytosol or import proteins that stabilize the DV membrane from the inner leaflet. A highly resolved DV proteome with and without heat stress could help with exploring this possibility.

Each year, millions of patients are prescribed antipyretics to treat malarial fever, despite evidence for contradictory clinical outcomes with this routine practice (*Brandts et al., 1997*; *Lell et al., 2001*).

A previous trial in Gabon showed that co-administrating naproxen (a nonsteroidal anti-inflammatory drug that relieves fever) and quinine did not interfere with parasite clearance by quinine (*Lell et al., 2001*). Another study using a monoclonal antibody against tumor necrosis factor (the primary mediator that induces fever) showed that fever reduction did not compromise parasite clearance by chloroquine in patients (*Kwiatkowski, 1993*). However, intravenous administration of the common nonsteroidal antipyretic ibuprofen in patients significantly delayed parasite clearance by artesunate-based combination therapy (*Krudsood et al., 2010*). Our study may provide a plausible explanation for these seemingly contradictory results: artesunate may reduce heat tolerance in *P. falciparum*, perhaps through altered PI(3)P levels (*Mbengue et al., 2015*). Therefore, alleviating heat stress by co-administrating antipyretics may prolong parasite clearance by artesunate in malaria patients. More clinical evidence will be required to determine if a causal link exists between malarial fever and parasite clearance when artemisinin-based combination therapy is employed.

Overall, we have shown that PI(3)P stabilizes the *P. falciparum* digestive vacuole and helps prevent heat-shock-induced parasite death. We investigated the parasite PI(3)P interactome and by utilizing chemical, biochemical and genetic approaches, identified PfHsp70-1 as a potential PI(3)P effector protein that maintains digestive vacuole integrity under heat stress. Our research highlights the role of PI(3)P in *Plasmodium* biology and lays the foundation for future work aimed at elucidating the parasite stress responses in greater mechanistic detail.

# Materials and methods

## *P. falciparum* culture

*P. falciparum* strains were cultured in complete medium (10.44 g/L RPMI 1640 (ThermoFisher Scientific), 25 mM HEPES (ThermoFisher Scientific), pH 7.2, 0.37 mM hypoxanthine (Sigma), 24 mM sodium bicarbonate (Sigma), 0.5% (wt/vol) AlbuMAX II (ThermoFisher Scientific), 25 µg/mL gentamicin (Sigma)) supplemented with freshly washed human RBCs (Gulf Coast Regional Blood Center, Houston, TX) every other day (*Radfar et al., 2009*). The parasite cultures were maintained at 2% parasitemia and 1% hematocrit at 37°C in a 3% $O_2$, 5% $CO_2$, 92% $N_2$ atmosphere. Synchronized cultures were obtained by treatment with 25 volumes of 5% (wt/vol) D-sorbitol (Sigma) at 37°C for 10 min during the early ring stage (<6 hr after reinvasion). The wild-type strain 3D7 (MRA-102) and PM2GT (MRA-805) (*Klemba et al., 2004*) were obtained from BEI Resources.

## Preparation of *P. falciparum* parasite protein extracts

Synchronized 3D7 parasite cultures (10–15% parasitemia and 1% hematocrit) at 38–44 hpi were pelleted at 300 *g*, resuspended in one volume of RPMI–sorbitol solution (10.44 g/L RPMI 1640, 25 mM HEPES, 24 mM sodium bicarbonate, 5% D-sorbitol) and layered on top of Percoll (Sigma) density gradients (6 mL 40% Percoll, 6 mL 60% Percoll, 6 mL 70% Percoll and 8 mL 80% Percoll in RPMI–sorbitol solution) (*Moll et al., 2013*). The gradients were centrifuged at 4300 *g* for 30 min at 20°C. Parasites in the 60–70% fraction were harvested and washed in 10 volumes of cold PBS at 800 *g* for 20 min at 4°C. The cells were resuspended in 10 volumes of cold 0.03% (wt/vol) saponin (Sigma) in PBS at 4°C for 15 min and centrifuged at 4300 *g* for 10 min at 4°C. The pellets were washed four times in 10 volumes of cold PBS at 4300 *g* for 10 min at 4°C and resuspended in four volumes of cold lysis/binding buffer (10 mM HEPES, pH 7.4, 150 mM NaCl (Fisher Chemical), 1 mM EDTA (Fisher Chemical), 0.25% (vol/vol) Triton X-100 (Fisher Chemical), 0.1% (vol/vol) 2-mercaptoethanol (MP Biomedicals), 1 mM benzamidine hydrochloride (Sigma), 1x complete EDTA-free protease inhibitor cocktail (Roche)). Parasite proteins were extracted by sonication at 35% amplitude for 10 s, six times with 1 min intervals on ice. The soluble fractions were collected by centrifuging at 20,000 *g* for 10 min at 4°C. Protein concentrations were determined using Pierce Coomassie Plus (Bradford) Assay Kit (ThermoFisher Scientific).

## PI(3)P bead-protein pull-down assay

Parasite protein extracts were pretreated with Benzonase nuclease (Sigma) (70 U to protein extract from 1 L culture at a final concentration of 90 U/mL) at 4°C for 30 min to reduce the interference of nucleic acids. To minimize non-specific binding proteins, 5 mg parasite protein lysate was pre-cleaned with 600 µL uncoated agarose beads (Echelon Biosciences, Salt Lake City, UT) at 4°C for 90

min. For each pull-down assay, 2.5 mg precleaned lysates at 4 mg/mL were incubated with 200 µL PI (3)P-conjugated beads or the control beads (Echelon Biosciences) at 4°C for 3 hr on a Labquake rotator (ThermoFisher Scientific). The beads were washed with 1 mL cold lysis/binding buffer four times at 300 $g$ for 4 min at 4°C. Proteins that bound to PI(3)P or control beads were eluted by boiling at 90–95°C for 20 min in 70 µL 4x Laemmli sample buffer (Bio-Rad) and centrifuging at 20,000 $g$ for 15 min. Proteins were resolved on Novex 4–20% tris–glycine gels (ThermoFisher Scientific) and visualized using a Pierce silver stain kit (ThermoFisher Scientific).

## Mass spectrometry

Each gel lane was sliced into 40–50 pieces, cut into small cubes (1–2 mm$^3$) and transferred to a 96-well plate with perforated wells. The silver-stained gel samples were destained in 50 µL freshly prepared mixture containing 15 mM potassium ferricyanide and 50 mM sodium thiosulfate for 10 min. The destained gels were washed in 50 µL of 25 mM ammonium bicarbonate, dehydrated with 50 µL acetonitrile for 5 min, and rinsed again with 50 µL acetonitrile. This wash–dehydration step was repeated twice more. In-gel digestion was performed by adding 25 µL of 4 µg/mL sequencing grade modified trypsin (Promega) in 25 mM ammonium bicarbonate, incubating for 10 min and adding additional 15 µL of 25 mM ammonium bicarbonate for overnight incubation. The peptide samples were harvested at 1000 rpm for 1 min. To increase extraction efficiency, the gels were further incubated with 40 µL acetonitrile containing 1% trifluoroacetic acid (TFA) for 5 min and centrifuged at 1000 rpm for 1 min. The pooled peptide extracts were lyophilized and resuspended in 5 µL of 50% acetonitrile and 0.5% TFA. Each sample was spotted onto a titanium dioxide-coated plate (0.15 µL/spot) and covered with 0.15 µL matrix mixture (a saturated solution of α-cyano-4-hydroxycinnamic acid (Sigma) in 50% acetonitrile, 0.825% TFA and 12.25 mM ammonium citrate) for matrix-assisted laser desorption/ionization–time of flight mass spectrometry (MALDI–TOF MS) analysis.

## Cloning of candidate genes

Total RNA was extracted from synchronized 3D7 parasites using the RNeasy Midi kit (Qiagen) and converted to cDNA. The genes of interest and *pfhsp70-1*$^{LID-}$ were PCR-amplified using specific primer pairs (*Supplementary file 1*) and Platinum *Taq* DNA polymerase (ThermoFisher Scientific) (2 U/reaction). The DNA fragment of the PI(3)P-specific binding probe 2xFyve was amplified from pEGFP-2xFYVE (*Gillooly, 2000*) (kindly provided by Harald Stenmark, Norwegian Radium Hospital, Oslo, Norway). The PCR products were subjected to agarose gel electrophoresis, followed by extraction using the Wizard SV gel and PCR clean-up system (Promega). The purified candidate genes were cloned into pCR8/GW/TOPO entry vectors (ThermoFisher Scientific) following the manufacturer's instruction. The sequence-verified open reading frames were subcloned into a pYES-DEST52 yeast expression vector (ThermoFisher Scientific) by the LR reaction. The purified plasmids harboring the genes of interest were transformed into yeast strain INV*Sc*1 (ThermoFisher Scientific) for protein expression and purification. Competent yeast cells were generated using the *Sc* Easy-Comp transformation kit (ThermoFisher Scientific) and the transformation was performed according to the manufacturer's instruction.

## Recombinant protein expression and purification

Each yeast transformant was inoculated into 5 mL SC–Ura medium (1.7 g/L yeast nitrogen base without amino acids, carbohydrate and ammonium sulfate (US Biological), 5 g/L ammonium sulfate (US Biological), 2 g/L drop-out mix synthetic minus uracil without yeast nitrogen base (US Biological)) containing 2% (wt/vol) glucose (Sigma) and incubated at 30°C for 24 hr with shaking (250 rpm). The cultures were expanded by adding 1.8 mL of the precultures to 2 L SC–Ura media containing 2% (wt/vol) raffinose (US Biological) and incubating overnight at 30°C at 250 rpm. Protein syntheses were induced by 2% (wt/vol) galactose (VWR International) at 30°C for 4 hr at 250 rpm when the OD$_{600}$ reached 0.8–1.0. Cells were then harvested by centrifugation at 4,000 $g$ for 10 min at 4°C, washed in 40 mL cold ultrapure water and immediately stored at −80°C. For protein purification, each pellet was thawed and mixed with one volume of 0.7 mm zirconia beads (Biospec Product, Germany) and 5–6 volumes of lysis buffer (50 mM NaH$_2$PO$_4$ (Fisher Chemical), pH 8.0, 300 mM NaCl, 10 mM imidazole (Fisher Chemical), 10% glycerol (VWR International), 0.1% Triton X-100) containing fresh protease inhibitors (0.1% 2-mercaptoethanol, 1 mM phenylmethylsulfonyl fluoride (PMSF;

ThermoFisher Scientific), 1 mM benzamidine hydrochloride, 1 mM Pefabloc SC (DSM Nutritional Products, Switzerland), 1x complete EDTA-free protease inhibitor cocktail). The cells were lysed by vigorous vortexing for 1 min, six times with 1 min intervals on ice, and the lysates were cleared by centrifugation at 4,300 $g$ for 15 min at 4°C. The lysates were then incubated with 1 mL prewashed Ni-NTA resin (Qiagen) at 4°C for 2 hr on a rotator. The protein-bound resins were washed in 25 mL wash buffer I (50 mM $NaH_2PO_4$, pH 8.0, 300 mM NaCl, 30 mM imidazole, 10% glycerol, 0.1% Triton X-100, 0.1% 2-mercaptoethanol, 1 mM PMSF) four times and 25 mL wash buffer II (20 mM HEPES, pH 8.0, 150 mM NaCl, 30 mM imidazole, 10% glycerol, 0.05% Triton X-100, 0.1% 2-mercaptoethanol, 1 mM PMSF) twice at 230 $g$ for 3 min at 4°C. Proteins were eluted with 5 mL elution buffer (50 mM HEPES, pH 7.5, 150 mM NaCl, 300 mM imidazole, 10% glycerol, 0.05% Triton X-100) three times at 4°C for 1 hr on a rotator, followed by centrifuging at 230 $g$ for 3 min at 4°C. Proteins were concentrated and buffer exchanged into the gel filtration buffer (50 mM Tris-HCl (Fisher Chemical), pH 7.5, 150 mM NaCl, 5% glycerol, 5 mM dithiothreitol (VWR International)) using Macrosep advance centrifugal device (Pall Corporation) with a molecular weight cut-off of 10 kDa. The proteins of interest were further purified using a Superdex 200 pg column on an NGC liquid chromatography system (Bio-Rad). Each fraction was resolved on Novex 4–20% tris–glycine gels with Coomassie blue staining to assess protein purity.

## Protein–lipid overlay assay

The lipid-binding capacities of the candidate proteins were assessed using a previously reported method (*Dowler et al., 2002*). Briefly, 1 µL of 100 µM and 500 µM PI (Echelon Biosciences) and PI(3) P diC 16 (Echelon Biosciences) in methanol:chloroform:$H_2O$ (2:1:0.8, vol/vol/vol) were spotted onto an Amersham Protran Supported 0.45 NC nitrocellulose membrane (GE healthcare) using a 10 µL 26s-gauge glass syringe (Hamilton, Reno, NV). The air-dried membranes were blocked with 3% fatty acid-free BSA (Millipore Sigma) in TBST (50 mM Tris-HCl, pH 7.5, 150 mM NaCl, 0.1% Tween 20) at room temperature for 2 hr. The membranes were separately incubated with 25–100 nM purified 6xHistidine-tagged proteins in the blocking buffer at 4°C for 16 hr with gentle shaking. After washing with TBST for 5 min four times, the membranes were probed with 1 µg/mL HisProbe–HRP conjugate (ThermoFisher Scientific) in TBST at room temperature for 1 hr. The membranes were subsequently washed with TBST for 5 min six times, and the lipid-binding proteins were detected using the Super-Signal West Femto Maximum Sensitivity Substrate kit (ThermoFisher Scientific) and a ChemiDoc MP imaging system (Bio-Rad). To determine the binding specificities, each purified protein (25 nM) in the blocking buffer (10 mM Tris-HCl, pH 8.0, 150 mM NaCl, 3% fatty acid-free BSA) was probed to preblocked PIP strips (Echelon Biosciences) and incubated at 4°C for 16 hr. The PIP strips were washed (10 mM Tris-HCl, pH 8.0, 150 mM NaCl, 0.05% Tween 20) at room temperature for 5 min three times and incubated with 1 µg/mL HisProbe–HRP conjugate in TBS (10 mM Tris-HCl, pH 8.0, 150 mM NaCl) at room temperature for 1 hr. After four additional washes, the binding signals were determined as described above.

## Construction of PfHsp70-1-mCherry parasites

*pfhsp70-1-mCherry* was subcloned into a *P. falciparum* expression vector, pfYC103 as previously reported (*Wagner et al., 2013*). The plasmid was purified using a Qiagen Maxi kit and resuspended in CytoMix (25 mM HEPES, pH 7.6, 2 mM EGTA (Sigma), 5 mM $MgCl_2$ (Fisher Scientific), 8.66 mM $K_2HPO_4$ (Fisher Scientific), 1.34 mM $KH_2PO_4$ (VWR International), 120 mM KCl (Fisher Scientific), 0.15 mM $CaCl_2$ (Sigma)) (*Rug and Maier, 2013*; *Crabb et al., 2004*). For transfection of *P. falciparum* 3D7, 100 µL ring-stage parasites (14–18 hpi) at 5% parasitemia was resuspended in 300 µL CytoMix containing 75 µg plasmid DNA in a 0.2 cm electroporation cuvette (Bio-Rad). Electroporation was performed at 0.31 kV and 950 µF with maximal capacitance using a Gene Pulser II system (Bio-Rad). The parasites were then cultured in complete medium at 1% hematocrit at 37°C. Selection of transfectants with 200 nM pyrimethamine (Sigma) started at 3 days post-transfection. PfHsp70-1-mCherry expression in transfected parasites was verified by fluorescence microscopy.

## Construction of PfHsp70-1 tunable line

CRISPR-Cas9 was used to modify the native *pfhsp70-1* (PF3D7_0818900) locus and install at the 3′ UTR key components of the TetR-DOZI-RNA aptamer system for conditional regulation of protein

expression (*Ganesan et al., 2016*). The donor vector used for modifying *pfhsp70-1* was made by Gibson assembly into the pSN054 vector (*Nasamu et al., 2019*) using DNA sequences summarized in *Supplementary file 1*. The right homology region (RHR) was PCR-amplified and inserted into pSN054 using the I-SceI restriction site. The left homology region (LHR; bp 1180–1752) fused to the re-codonized 3′ end of the gene (bp 1753–2034) and the target-specifying single guide RNA were synthesized on the BioXP 3200 (SGI-DNA) and cloned into pSN054 using the FseI/AsisI and AflII restriction sites, respectively. pSN054 encodes Blasticidin S-deaminase for selecting transgenic parasites and the reporter gene Renilla luciferase (*RLuc*). The final *pfhsp70-1*_pSN054 construct was confirmed by restriction digest mapping and Sanger sequencing. Transfection into *Sp*Cas9- and T7 RNA polymerase-expressing NF54 parasites (*Wagner et al., 2014*) was carried out by preloading erythrocytes with the *pfhsp70-1*_pSN054 plasmid as described previously (*Deitsch et al., 2001*). Cultures were maintained in 500 nM aTc (Sigma) and 2.5 µg/mL Blasticidin S hydrochloride (RPI Corp). Parasitemia was monitored using Giemsa-stained smears and RLuc measurements. Clonal parasites were obtained by limiting dilution (*Rosario, 1981*).

## *P. falciparum* growth assays

Synchronized 3D7 parasites at 2% parasitemia and 1% hematocrit were grown to 28–37 hpi. To assess the effect of febrile temperature on *P. falciparum* growth, 1 mL parasite cultures were incubated at 40°C for 0–12 hr in triplicate, followed by incubating at 37°C for 45 hr. The cells were pelleted at 300 *g*, resuspended in 150 µL lysis buffer (20 mM Tris-HCl, pH 7.5, 5 mM EDTA dipotassium salt dihydrate, 0.0008% saponin, 0.001% Triton X-100) containing 500 ng/mL DAPI (ThermoFisher Scientific) and transferred to a 96-well black plate (Corning) (*Baniecki et al., 2007*). After incubation in the dark at room temperature for 30 min, the fluorescence signals were measured at 460 nm with excitation at 358 nm using a Victor X2 plate reader (PerkinElmer). Relative parasite loads were determined by subtracting the initial cell input signal from the heat-treated cell signals, followed by normalizing each signal intensity to a non-heat-treated control [($F_{x \text{ hours heat shock}} - F_{input}$)/($F_{no \text{ heat shock}} - F_{input}$)]. To monitor parasite development, blood smears were fixed in methanol and stained with 20% Giemsa solution (Ricca Chemical Company). Automatic image capturing was carried out using a Zeiss Axio imager Z2 widefield microscope (Carl Zeiss, Germany) and the parasite sizes were determined using ImageJ/FIJI (*Schindelin et al., 2012*). More than 300 parasites from each duplicate smear were measured.

To test if reduced PI(3)P levels inhibit parasite growth under the febrile condition, parasites at 32 hpi and 38 hpi were cultured at 37°C or 40°C for 6 hr in the presence or absence of 20 µM Wortmannin (Selleckchem, Houston, TX) or 40 µM LY294002 (Sigma). The cells were subsequently washed and resuspended in complete medium (1% hematocrit), followed by incubating at 37°C until 34 hr after reinvasion for the DAPI-based growth assay. For LY294002 treatment, parasites were maintained in the inhibitor-containing medium after heat shock to avoid parasite recovery caused by reversible PI3K inhibition.

To assess the drug hypersensitivities under heat shock, 100 µL of *P. falciparum* 3D7 parasite culture (2% parasitemia and 2% hematocrit) at the ring (10 hpi) and trophozoite (32 hpi) stages was dispensed into each well of two 96-well black plates containing 100 µL complete media with serial diluted Wortmannin, LY294002, LY303511 (Ark Pharm, Arlington Heights, IL), lapachol (Sigma), atovaquone (Sigma), pyrimethamine or quinacrine dihydrochloride (Sigma) in triplicate. For the PfHsp70-1 tunable line and the control YFP line, 100 µL culture (2% parasitemia and 2% hematocrit) at 10 hpi was dispensed into each well of two 96-well black plates containing 100 µL complete medium with 2.5 µg/mL Blasticidin S and 0–1 µM aTc in triplicate. Each well contained 0.5% DMSO. Plates were incubated at 37°C in a 3% $O_2$, 5% $CO_2$, 92% $N_2$ atmosphere for 48 hr and 72 hr with or without a 6 hr heat shock (32–38 hpi). At 34 hr post-reinvasion, 40 µL lysis solution (20 mM Tris-HCl, pH 7.5, 5 mM EDTA, 0.16% saponin, 1.6% Triton X-100) containing 10x SYBR Green I (ThermoFisher Scientific) was added to each well and incubated in the dark for 24 hr (*Kato et al., 2016*). The fluorescent signals were measured at 535 nm with excitation at 485 nm using an EnVision 2105 multimode plate reader (PerkinElmer). $EC_{50}$ determinations were completed using Prism (GraphPad Software).

To determine the viability of *P. falciparum* upon PfHsp70-1 knockdown, synchronized ring-stage parasites were cultured in the presence (3 nM and 50 nM) or absence of aTc in triplicate in 96-well U-bottom plates (Corning). Luminescence was measured at 0, 72 and 120 hr post-treatment using

the Renilla-Glo luciferase assay system (Promega) and a GloMax Discover multimode microplate reader (Promega). Luminescence values were normalized to chloroquine-treated (200 nM) parasites and analyzed using Prism.

## Drug sensitivity assay for the PfHsp70-1 tunable line

The tunable PfHsp70-1 and the control YFP parasite lines at the ring stage were resuspended in media containing aTc at varying concentrations (0–50 nM) and dispensed into each well of 96-well U-bottom plates containing serial diluted Wortmannin (0–640 µM), LY294002 (0–200 µM) or Bafilomycin A (0–100 nM). DMSO and chloroquine (200 nM) were negative and positive controls, respectively. Luminescence was measured at 72 hr post-treatment to determine $EC_{50}$ values using Prism.

## PI(3)P detection in *P. falciparum* lipid extracts

*P. falciparum* 3D7 parasites (32 hpi) at 30–40% parasitemia and 0.5% hematocrit were cultured at 37°C or 40°C for 6 hr. Parasites were pelleted at 400 $g$ for 10 min and incubated in 10 volumes of cold 0.03% saponin in PBS at 4°C for 15 min. After centrifugation at 4300 $g$ for 10 min at 4°C, the pellets were washed in 30 mL PBS at 4300 $g$ for 10 min, resuspended in 3 mL methanol:chloroform (2:1, vol/vol), and transferred to Pyrex glass conical tubes (Sigma). Parasite lipids were extracted by sonication. The lipids were further solubilized by vortexing every 5 min for 30 min. An additional 500 µL chloroform and 900 µL ultrapure water were added to the mixture, followed by vortexing and centrifuging at 1300 $g$ for 10 min. The bottom organic layers were transferred to glass tubes (100 × 13 mm) with Teflon-lined caps (Thomas Scientific), mixed with 500 µL chloroform and 900 µL ultrapure water, and centrifuged at 1300 $g$ for 10 min. The bottom organic layers were transferred to clean glass vials, followed by evaporation under nitrogen to obtain dried lipid films. The extracted lipids were dissolved in methanol:chloroform (1:1, vol/vol) at 25–30 mg/mL for the protein–lipid overlay assay. As described above, 1 µL each lipid extract was spotted onto a nitrocellulose membrane in triplicate, followed by blocking in 3% fatty acid-free BSA in TBST. The membranes were probed with 500 nM purified 6xHistidine-tagged PI(3)P-specific binding peptide 2xFyve in the blocking buffer at 4°C for 16 hr. The bound 2xFyve was detected using 1 µg/mL HisProbe–HRP conjugate in TBS and electrochemiluminescence. Images were analyzed by ImageJ/FIJI (*Schindelin et al., 2012*) to quantify relative PI(3)P levels.

## Live cell microscopy

*Plasmodium* 3D7 parasites at 32 hpi (2% parasitemia and 0.5% hematocrit) were incubated in complete media containing 100 µM LysoTracker Red DND-99 (ThermoFisher Scientific) and 20 µM Wortmannin, 40 µM LY294002, 40 µM LY303511, 40 µM lapachol, 40 nM atovaquone, 200 nM pyrimethamine, 100 µM bortezomib (Selleckchem), 1 µM 15-deoxyspergualin trihydrochloride (Toronto Research Chemicals) or 20 nM artesunate in the dark at 37°C or 40°C for 6 hr. DMSO was used as a negative control at a final concentration of 0.032–0.2%. The dye-loaded cells were washed in 5 mL complete medium at 400 $g$ for 3 min and resuspended in media containing the corresponding inhibitors. Live parasites were applied to a Nunc 8-well Lab-Tek chambered coverglass (ThermoFisher Scientific) (200 µL/well) and immediately imaged using a Zeiss LSM 880 inverted confocal microscope with Airyscan (Carl Zeiss, Germany). More than 20 images were taken per well with the same microscopic system and settings. To minimize laser-induced photolysis of the *P. falciparum* DV membrane (*Wissing et al., 2002*; *Rohrbach et al., 2005*), the microscope settings for LysoTracker Red-loaded parasites were optimized as follows: (1) a 561 nm diode laser was used for excitation with 4% transmission and a detector gain of 800 V; (2) a 1024 × 1024 pixel scan with a 0.38 µsec pixel time (1.89 s/image) was applied; (3) the pixel averaging was set to 2; (4) single images were obtained using a 63x oil-immersion objective lens (Plan-Apochromat, NA 1.4) with a two-fold zoom. Recovery of DV membranes was further assessed by incubating heat-shocked parasites in the dark at 37°C for additional 3 hr with or without LY294002.

For the PfHsp70-1 tunable line, parasites (2% parasitemia and 0.5% hematocrit) were pre-cultured in complete media containing 2.5 µg/mL Blasticidin S and 50 nM or 500 nM aTc at 37°C for 24 hr before heat shock and LysoTracker Red loading. For strain PM2GT without LysoTracker Red loading, the experimental conditions remained the same with the following modifications: (1) parasites were

treated for 5 hr; (2) a 488 nm Argon/2 laser was used for excitation with 5% transmission and a detector gain of 800 V.

## Mitochondrial membrane potential assay

Parasites after treatment were washed in 5 mL complete medium at 400 $g$ for 3 min and stained with 5 µM JC-1 (ThermoFisher Scientific) in complete medium in the dark at 37°C for 30 min. Cells were washed once and applied to a Nunc 8-well Lab-Tek chambered coverglass for microscopy. Live-cell imaging was performed as described above with the following modifications: (1) a 488 nm Argon/2 laser was used for excitation with 5% transmission; (2) green fluorescence from monomeric JC-1 in the cytoplasm was detected using an emission filter of 516–553 nm with a detector gain of 650 V; (3) red fluorescence from JC-1 aggregates in active mitochondria was detected using an emission filter of 580–610 nm with a detector gain of 480 V. Parasites treated with 50 µM carbonyl cyanide 3-chlorophenylhydrazone (CCCP) (Sigma) at 37°C for 1 hr were used as a positive control for mitochondrial depolarization. DMSO was used as a negative control at a final concentration of 0.08%. More than 20 parasites were imaged per well with the same microscopic system and settings.

## Image analysis

All confocal images were exported as TIF files at the same brightness and contrast settings with no gamma correction using ZEN 2.3 software (Carl Zeiss, Germany). Exported images were analyzed by ImageJ/FIJI (*Schindelin et al., 2012*). The subcellular locations of hemozoin crystals (approximate DV area) in *P. falciparum* were determined by DIC images using the thresholding algorithm, Intermodes (*Prewitt and Mendelsohn, 1966*) in ImageJ/FIJI. The mean fluorescence intensities (MFIs) of the DVs (hemozoin areas) were measured after background subtraction. For the strain PM2GT, the overall parasite MFIs and the DV-excluded areas (whole parasite – DV) were measured as well. The parasite areas were determined in GFP channel using the thresholding algorithm, Percentile (*Doyle, 1962*). The DV-excluded areas were determined using XOR function in ImageJ/FIJI. For JC-1 assays, images from green and red channels were exported at the same brightness and contrast. Fluorescence intensities in both channels were quantified using ImageJ/FIJI to calculate the ratios of JC-1 red/green.

## Western blot analysis

*P. falciparum* 3D7 parasites (32 hpi) at 10–15% parasitemia and 1% hematocrit were cultured in 10 mL complete media containing 20 µM Wortmannin, 40 µM LY294002, 40 µM LY303511 or 0.08% DMSO at 37°C or 40°C for 6 hr. Cells were washed in 10 mL PBS and treated with 4 mL 0.03% saponin in PBS at 4°C for 15 min. The parasites were pelleted and washed three times in 10 mL cold PBS at 4300 $g$ for 10 min at 4°C to remove host erythrocyte proteins, followed by incubating in 200 µL lysis buffer (4% (wt/vol) sodium dodecyl sulfate (VWR International), 0.5% Triton X-100 in PBS) containing 5–10 U DNase I (Zymo Research) for 30 min at room temperature. Parasite proteins were harvested by centrifuging at 20,000 $g$ for 10 min, resolved on Novex 4–20% tris–glycine gels and transferred to nitrocellulose membranes using the Trans-Blot Turbo Transfer system (Bio-Rad). As a loading control, the membranes were stained with Ponceau S (Sigma) for 5 min, rinsed and imaged using the ChemiDoc MP imaging system. Membranes were then destained with 0.1 M NaOH for 1 min and rinsed with ultrapure water. After blocking with 3% BSA in PBS for 1 hr, membranes were probed with a rabbit K48-linkage specific ubiquitin antibody (Abcam) at a 1:1000 dilution overnight at 4°C. After three washes in 10 mL PBS containing 0.2% Tween 20 (PBST) for 5 min, membranes were incubated with Alex Fluor 488-conjugated chicken anti-rabbit IgG (H+L) antibody (ThermoFisher Scientific) at a 1:1000 dilution in the dark for 1 hr. The membranes were washed four times in 10 mL PBST for 5 min before imaging. The images were analyzed by ImageJ/FIJI (*Schindelin et al., 2012*). The overall K48-ubiquitin signals were normalized to the corresponding Ponceau S signals. For PfHsp70-1 quantification in the wild-type 3D7 and PfHsp70-1 strains, a rabbit PfHsp70-1 antibody (QED Bioscience) at a 1:1000 dilution was applied.

## Acknowledgements

This work was supported by the National Institutes of Health (NIH) (DP2AI138239 to ERD) and the Bill and Melinda Gates Foundation (BMGF) (OPP1132312 and OPP1162467 to JCN). The content of this study is solely the responsibility of the authors and does not necessarily represent the official views of the NIH. The following parasite strains were obtained through Biodefense and Emerging Infections Research Resources Repository, National Institute of Allergy and Infectious Diseases, NIH: *Plasmodium falciparum*, strain 3D7, MRA-102, contributed by Daniel J Carucci and *Plasmodium falciparum*, strain PM2GT, MRA-805, contributed by Daniel E Goldberg. We are grateful to Timothy Haystead for help with mass spectrometry, to the Duke Light Microscopy Core Facilities and the Derbyshire lab for critical reading of this manuscript. We thank Harald Stenmark for providing the pEGFP-2xFYVE plasmid.

## Additional information

### Funding

| Funder | Grant reference number | Author |
|---|---|---|
| National Institutes of Health | DP2AI138239 | Emily Derbyshire |
| Bill and Melinda Gates Foundation | OPP1132312 | Jacquin C Niles |
| Bill and Melinda Gates Foundation | OPP1162467 | Jacquin C Niles |

The funders had no role in study design, data collection and interpretation, or the decision to submit the work for publication.

### Author contributions

Kuan-Yi Lu, Conceptualization, Formal analysis, Investigation, Visualization, Methodology, Writing - original draft, Writing - review and editing; Charisse Flerida A Pasaje, Formal analysis, Investigation, Visualization, Writing - review and editing; Tamanna Srivastava, Investigation, Writing - review and editing; David R Loiselle, Formal analysis, Investigation, Writing - review and editing; Jacquin C Niles, Resources, Formal analysis, Supervision, Methodology, Writing - original draft; Emily Derbyshire, Conceptualization, Resources, Supervision, Funding acquisition, Writing - review and editing

### Author ORCIDs

Kuan-Yi Lu https://orcid.org/0000-0002-3663-377X
Charisse Flerida A Pasaje http://orcid.org/0000-0002-9780-3680
David R Loiselle http://orcid.org/0000-0002-7065-8495
Jacquin C Niles http://orcid.org/0000-0002-6250-8796
Emily Derbyshire https://orcid.org/0000-0001-6664-8844

### Decision letter and Author response

Decision letter https://doi.org/10.7554/eLife.56773.sa1
Author response https://doi.org/10.7554/eLife.56773.sa2

## Additional files

### Supplementary files

- Supplementary file 1. Primers for gene cloning.
- Supplementary file 2. Reported EC50 values of small molecule inhibitors used in this study.
- Transparent reporting form

### Data availability

All data generated or analysed during this study are included in the manuscript and supporting files.

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
