## [Decision Letter]

**Acceptance summary:**

The study proposes an important link between PI(3)P levels and HSP70-1 in protecting *Plasmodium falciparum* from heat shock (HS). The authors used a wide variety of complementary techniques and controls to show that digestive vacuolar membrane integrity in *Plasmodium falciparum* is linked to surviving clinically-relevant heat shock. This work may be relevant for the use of antipyretics and therapies that interrupt this pathway, such as drugs contained in artemisinin-based combination therapies, and may therefore have an impact on the treatment of malaria.

**Decision letter after peer review:**

Thank you for submitting your article "Phosphatidylinositol 3-phosphate and Hsp70 protect *Plasmodium falciparum* from heat-induced cell death" for consideration by *eLife*. Your article has been reviewed by two peer reviewers, one of whom is a member of our Board of Reviewing Editors, and the evaluation has been overseen Dominique Soldati-Favre as the Senior Editor. The reviewers have opted to remain anonymous.

The reviewers have discussed the reviews with one another, and the Reviewing Editor has drafted this decision to help you prepare a revised submission.

Summary:

The study proposes an important link between PI(3)P levels and HSP70-1 in protecting *Plasmodium falciparum* from heat shock (HS). The authors used a wide variety of complementary techniques and controls to show that digestive vacuolar membrane integrity in *Plasmodium falciparum* is linked to surviving clinically-relevant heat shock. This may be relevant for the use of antipyretics and therapies that interrupt this pathway, such as drugs contained in artemisinin-based combination therapies. The authors rely strongly on available inhibitors of PI3K (Wortmanin and LY294) to create a state in which PI(3)P is depleted, which renders parasites more susceptible to HS. Parasite death is related to a loss of digestive vacuole integrity, which is measured by two fluorescence-based assays that largely agree with one another-compounds that synergize with HS, also appear to compromise the integrity of the digestive vacuole. The connection between the PI(3)P and HSP70-1 is less clearly defined. Assays for lipid binding are insufficiently controlled, as is the evidence for synergy between PI(3)P and HSP70-1. Clearly establishing this interaction will be critical to support the thesis of the manuscript. Some attempts to further elucidate the mechanism were unsuccessful (e.g., unchanged K48 ubiquitination), but I consider that beyond the scope of this publication.

Essential revisions:

1) References for the effects of PI3K inhibitors do not cite primary literature. Citation 16 references two primary reports: one which uses Wortmanin (100 nM) for 1.5 hours, and another that uses both Wortmanin and LY294 but does not include the LY303 nor performs measurements of PI(3)P. The effect of Wortmanin and LY294 (but not LY303) on parasite PI(3)P levels therefore needs to be established within the experimental system, and should be monitored against an invariant lipid. If possible, cholesterol would also be interesting to measure because of its well-known role in temperature and membrane fluidity.

2) Binding of recombinant proteins to lipids is provided with no negative control. Moreover, the follow-up analysis of the HSP70-1 domain necessary for binding is provided on a different assay, so there's no ability to control Figure 6. Equal binding of all phosphatidylinositol monophosphates is of particular concern, hinting at a potential issue with the assay. Ideally, a second measure of lipid binding (e.g. liposome-based assays) would be provided with the appropriate controls.

3) The hypersensitivity to heat shock of the conditional HSP70-1 strain casts some doubt on the results, raising the possibility that different *P. falciparum* strains respond differently to the assay. There are a few options to deal with this issue. The authors could demonstrate that the strain is indeed hypomorphic by measuring endogenous PI(3)P levels, compared to wild-type, for example by probing cell extracts with 2xFyve peptide. The authors could alternatively show that the strain from which the conditional was derived is not intrinsically HS sensitive, demonstrating that the defect was introduced in the final manipulation. Complementation would be ideal but is not required as it is not standard for the Plasmodium field; however, if attempted, the comparison between wild-type HSP70 and the LID mutant. The authors could also test a range of heat-shock treatments, as performed in Figure 1—figure supplement 1, to establish conditions that reveal any difference in the HS response between the presence and absence of aTc. We do not expect all of these experiments to be performed, but are trying to provide examples of the type of data that would strengthen the link between PI(3)P, HSP70-1, and the heat-shock response.

4) At different points the authors use various control compounds to contrast their effects to those of the PI3K inhibitors. It is not explained why the set of controls used is different between experiments: (Figure 1—figure supplement 3) atovaquone, pyrimethamine, quinacrine, and lapachol; (Figure 3—figure supplement 1) atovaquone, pyrimethamine, and lapachol; (Figure 7—figure supplement 6) Bafilomycin A. A clear rationale for the choice of control compounds should be provided. If other control compounds were used but did not give the expected outcome, these should be reported.

---

## [Author Response]

Summary:The study proposes an important link between PI(3)P levels and HSP70-1 in protecting *Plasmodium falciparum* from heat shock (HS). The authors used a wide variety of complementary techniques and controls to show that digestive vacuolar membrane integrity in Plasmodium falciparum is linked to surviving clinically-relevant heat shock. This may be relevant for the use of antipyretics and therapies that interrupt this pathway, such as drugs contained in artemisinin-based combination therapies. The authors rely strongly on available inhibitors of PI3K (Wortmanin and LY294) to create a state in which PI(3)P is depleted, which renders parasites more susceptible to HS. Parasite death is related to a loss of digestive vacuole integrity, which is measured by two fluorescence-based assays that largely agree with one another-compounds that synergize with HS, also appear to compromise the integrity of the digestive vacuole. The connection between the PI(3)P and HSP70-1 is less clearly defined. Assays for lipid binding are insufficiently controlled, as is the evidence for synergy between PI(3)P and HSP70-1. Clearly establishing this interaction will be critical to support the thesis of the manuscript. Some attempts to further elucidate the mechanism were unsuccessful (e.g., unchanged K48 ubiquitination), but I consider that beyond the scope of this publication.Essential revisions:1) References for the effects of PI3K inhibitors do not cite primary literature. Citation 16 references two primary reports: one which uses Wortmanin (100 nM) for 1.5 hours, and another that uses both Wortmanin and LY294 but does not include the LY303 nor performs measurements of PI(3)P. The effect of Wortmanin and LY294 (but not LY303) on parasite PI(3)P levels therefore needs to be established within the experimental system, and should be monitored against an invariant lipid. If possible, cholesterol would also be interesting to measure because of its well-known role in temperature and membrane fluidity.

Wortmannin and LY294002 have been extensively used as PI3K inhibitors in *Plasmodium* and *Toxoplasma* (a closely related parasite) (Mbengue et al., 2015; Tawk et al., 2010; Bansal et al., 2017; Dalal and Klemba, 2015; Tawk et al., 2011; Kitamura et al., 2012; Besteiro et al., 2011; Stutz et al., 2012). Biochemical studies have shown that Wortmannin reduces the level of PI(3)P, but not other phosphoinositides, in *P. falciparum* (Tawk et al., 2010). Treatments with Wortmannin and LY294002, but not the inactive analog LY303511, blocked PI(3)P production as revealed by a PI(3)P-specific probe (Mbengue et al., 2015). Our study used these compounds at comparable concentrations and conditions to probe the importance of PI(3)P. Furthermore, we tested a range of antimalarial drugs with various modes of action and only those known to inhibit PI(3)P synthesis (Wortmannin, LY294002 and artesunate) showed the observed phenotype. While we agree a study of the role of cholesterol in the heat response would be interesting, we believe it is outside the scope of the current study. We have checked the manuscript to ensure that our language is clear about our goal to specifically evaluate the functional role of PI(3)P in *P. falciparum* under heat stress and added additional references to previous studies using the PI3K inhibitors.

2) Binding of recombinant proteins to lipids is provided with no negative control. Moreover, the follow-up analysis of the HSP70-1 domain necessary for binding is provided on a different assay, so there's no ability to control Figure 6. Equal binding of all phosphatidylinositol monophosphates is of particular concern, hinting at a potential issue with the assay. Ideally, a second measure of lipid binding (e.g. liposome-based assays) would be provided with the appropriate controls.

We thank the reviewer for the suggestion to add a negative control for our lipid binding assay. We repeated our experiments with PfHsp70-1, PfRan, PfAlba1, and 2xFyve, and included a His-tagged non-candidate protein PfHop as a negative control (added to Figure 6). These experiments were repeated in duplicate. In contrast to the PI(3)P-binding proteins, PfHop did not bind to PI(3)P under the same assay condition. In this study, we employed the lipid dot blot assays and the commercial PIP strips with different conditions to validate the identified PI(3)P–protein interactions. These methods have been extensively used to identify lipid–protein interactions (Barneda et al., 2015; Elwell et al., 2017; Botero et al., 2019; He et al., 2017; Liu et al., 2016; Barnett et al., 2019; Liao et al., 2019; Zhang et al., 2010; Narayanan et al., 2018). While other techniques exist to determine the binding coefficients to different lipids, we believe that our approach validates the PI(3)P-binding capability identified from our pull-down study. Binding to PI(3)P were consistently observed using the three different platforms (chemoproteomics, commercial PIP strips, lipid dot blots). Our data show that (1) PfHsp70-1 binds to PI(3)P, (2) PfHsp70-1 does not bind to or has lower binding affinity to some other lipids, and (3) the binding is not simply a result of electrostatic association as PfHsp70-1 has lower affinity to phosphatidylinositol triphosphate, phosphatidylserine and some phosphatidylinositol bisphosphates. It is not uncommon that a protein can bind to multiple biomolecules (e.g., lipids). In fact, mammalian homologs of PfHsp70-1 can bind to various lipids, including different phosphatidylinositol monophosphates (Morozova et al., 2016; McCallister et al., 2016; McCallister et al., 2016; McCallister et al., 2015). Notably, PI(5)P and PI(3,5)P_2_ (that were associated with PfHsp70-1 in vitro) were not found in *Plasmodium* parasites (Tawk et al., 2010). We have added references and text to clarify these points. Importantly, through chemical inhibition, genetic and chemical–genetic approaches, we have demonstrated a functional association between PI(3)P and PfHsp70-1.

3) The hypersensitivity to heat shock of the conditional HSP70-1 strain casts some doubt on the results, raising the possibility that different *P. falciparum* strains respond differently to the assay. There are a few options to deal with this issue. The authors could demonstrate that the strain is indeed hypomorphic by measuring endogenous PI(3)P levels, compared to wild-type, for example by probing cell extracts with 2xFyve peptide. The authors could alternatively show that the strain from which the conditional was derived is not intrinsically HS sensitive, demonstrating that the defect was introduced in the final manipulation. Complementation would be ideal but is not required as it is not standard for the Plasmodium field; however, if attempted, the comparison between wild-type HSP70 and the LID mutant. The authors could also test a range of heat-shock treatments, as performed in Figure 1—figure supplement 1, to establish conditions that reveal any difference in the HS response between the presence and absence of aTc. We do not expect all of these experiments to be performed, but are trying to provide examples of the type of data that would strengthen the link between PI(3)P, HSP70-1, and the heat-shock response.

The reviewer suggests helpful control experiments to strengthen the link between PI(3)P, HSP70-1, and the heat-shock response. We carried out four sets of experiments to show that (1) the PfHsp70-1 strain was hypomorphic, (2) the hypersensitivity to both PI3K inhibitors was detected in the PfHsp70-1 strain but not in the control strain, and (3) the observed PI(3)P functions were conserved in the PfHsp70-1 strain. First, we examined if heat shock affects the parasite loads of the control line in which yellow fluorescent protein (YFP) expression is regulated by the same TetR-DOZI–aptamer-based system. Our data show that parasite loads in the YFP strain were not affected at various anhydrotetracycline (aTc) concentrations (0–1 μM) under the same stress condition (40 °C for 6 h) (added as Figure 7—figure supplement 4C). This indicates that the heat-sensitive phenotype of the PfHsp70-1 strain was not caused by aTc treatment or a general artifact of the genetic approach. Next, we compared the PfHsp70-1 protein level in the PfHsp70-1 line and the wild-type strain 3D7 using Western blot. Consistent with previous reports, heat shock induces PfHsp70-1 expression in *P. falciparum* 3D7 (added as Figure 7—figure supplement 5). Importantly, PfHsp70-1 expression was attenuated in the PfHsp70-1 conditional knockdown strain (as normally cultured in the complete medium with 500 nM aTc) at both 37 °C and 40 °C (Figure 7—figure supplement 5). Thus, the increased heat sensitivity and the heat shock-induced DV destabilization of the PfHsp70-1 line are likely due to the hypomorphic expression of PfHsp70-1. Additionally, we repeated the hypersensitivity assays with both PI3K inhibitors (Wortmannin and LY294002) in the PfHsp70-1 and control YFP strains. We consistently observed that downregulation of PfHsp70-1 induced hypersensitivity to both PI3K inhibitors. In contrast, downregulation of YFP expression using the same technique did not affect the activity of the PI3K inhibitors (added as Figure 7—figure supplement 8C). Finally, we tested if the different strains respond differently to the PI(3)P inhibitor treatments. Our data show that treatment with PI3K inhibitors sensitized heat shock-induced cell death in the PfHsp70-1 line, consistent with the observations using the wild-type 3D7 strain (added as Figure 7—figure supplement 6). As expected, heat-induced drug hypersensitivity was not detected with control compounds (LY303511 and atovaquone) (Figure 7—figure supplement 6). Together, these data show that PfHsp70-1 downregulation reduces parasite fitness and causes the DV destabilization under heat shock. We appreciate the reviewer’s suggestion to confirm that these phenotypes were not restricted to the wild-type 3D7 strain and were not an artifact of the TetR–aptamer-based system.

4) At different point the authors use various control compounds to contrast their effects to those of the PI3K inhibitors. It is not explained why the set of controls used is different between experiments: (Figure 1—figure supplement 3) atovaquone, pyrimethamine, quinacrine, and lapachol; (Figure 3—figure supplement 1) atovaquone, pyrimethamine, and lapachol; (Figure 7—figure supplement 6) Bafilomycin A. A clear rationale for the choice of control compounds should be provided. If other control compounds were used but did not give the expected outcome, these should be reported.

Various antimalarial drugs having different modes of action were selected based on availability and priority when the experiments in the noted figures were performed. Treatment with two different PI3K inhibitors reduced parasite fitness under heat shock, while none of the five control compounds showed heat shock-induced drug hypersensitivity (Figure 1 and Figure 1—figure supplement 3). We have provided rationale for the controls including their different modes of action. We then employed live cell confocal microscopy to determine the importance of PI(3)P in the *Plasmodium* digestive vacuole stability. These assays are laborious and costly; thus, four of the five control compounds were prioritized for these experiments. Again, we observed that the controls did not affect DV stability like the PI3K inhibitors did under the same experimental setting (three biological replicates with >20 parasites examined for each treatment and replicate; Figure 3 and Figure 3—figure supplement 1). In Figure 7—figure supplement 6, we felt that one compound (bafilomycin A) is sufficient to determine if the genetic modification and PfHsp70-1 knockdown caused a general drug hypersensitivity by increasing membrane permeability to drugs. We strategically selected bafilomycin A for this study since it inhibits the V-type ATPase complex, which localizes to membranes, including the DV. Therefore, the lack of differential hypersensitivity to bafilomycin A further suggests that PfHsp70-1 interacts with PI(3)P instead of a random DV component. We have clarified the use of our controls throughout the manuscript and confirmed that all treatments and assays were reported.